# CoSA: Compressed Sensing-Based Adaptation of Large Language Models

## Abstract

Parameter-Efficient Fine-Tuning (PEFT) has emerged as a practical paradigm for adapting large language models (LLMs) without updating all parameters. Most existing approaches, such as LoRA and PiSSA, rely on low-rank decompositions of weight updates. However, the low-rank assumption may restrict expressivity, particularly in task-specific adaptation scenarios where singular values are distributed relatively uniformly. To address this limitation, we propose **CoSA** (*Compressed Sensing-Based Adaptation*), a new PEFT method extended from compressed sensing theory. Instead of constraining weight updates to a low-rank subspace, CoSA expresses them through fixed random projection matrices and a compact learnable core. We provide a formal theoretical analysis of CoSA as a synthesis process, proving that weight updates can be compactly encoded into a low-dimensional space and mapped back through random projections. Extensive experimental results suggest that CoSA provides a principled perspective for efficient and expressive multi-scale model adaptation. Specifically, we evaluate CoSA on 10 diverse tasks including natural language understanding and generation, employing 5 models of different scales from RoBERTa, Llama, and Qwen families. Across these settings, CoSA consistently matches or outperforms state-of-the-art PEFT baselines while requiring over 68.4% fewer trainable parameters than LoRA and PiSSA.

## 1 Introduction

Pre-trained large language models (LLMs) (Vaswani et al., 2017; Touvron et al., 2023; Team et al., 2023; Liu et al., 2024a) have demonstrated exceptional performance across a wide spectrum of natural language processing (NLP) tasks (Nie et al., 2019; Gatt & Krahmer, 2018; Zeng et al., 2023). However, the adaptation of these models via full fine-tuning is computationally prohibitive, demanding extensive memory and processing resources (Xia et al.; Touvron et al., 2023; Chen et al., 2023). In response to this challenge, Low-Rank Adaptation (LoRA) methods as a popular Parameter-Efficient Fine-Tuning (PEFT) (Hu et al., 2022; Hayou et al., 2024; Zhang et al., 2023; Wang et al., 2024; Meng et al., 2024) method have emerged, which only update a small fraction of the model's parameters while keeping the vast majority of pre-trained weights frozen, thereby achieving performance comparable to full fine-tuning with significantly reduced resource consumption.

Despite their success, LoRA frameworks share a key limitation: they impose an explicit low-rank constraint on the weight update $\Delta W$. While this design achieves computational efficiency, it also imposes a rigid structural assumption that may not adequately represent the true geometry of task-specific updates. In practice, the optimal adaptation of $\Delta W$ can be distributed in many directions, making it poorly approximated by a restricted set of directions in the parameter space. Consequently, these methods are prone to approximation errors that limit expressivity and can degrade downstream performance (Hameed et al., 2024).

In contrast, we take a different perspective. Instead of constraining $\Delta W$ to a low-rank subspace, we posit that effective updates can be compactly represented within a task-agnostic basis defined by two fixed random projection matrices. This formulation allows adaptations to span diverse directions in parameter space without the rigid bottleneck of low-rank parameterizations. Evidence for this view comes from intrinsic dimensionality studies (Camastra & Staiano, 2016; Levina & Bickel, 2004; Ansuini et al., 2019), which reveal that fine-tuning operates in a surprisingly small subspace

of the full parameter space (Aghajanyan et al., 2020). If such a subspace can be stably accessed via random projections, adaptation reduces to estimating only a compact set of coefficients in the projected space (see Section 3.2). This design both lowers the number of trainable parameters and enhances robustness and accuracy in large-scale models.

Our proposed method, **Co**mpressed **S**ensing–based **A**daptation (CoSA), is motivated by the principle that high-dimensional signals can be represented compactly and stably using random projections. However, translating this idea to PEFT presents two challenges: (1) how to define a task-agnostic representation that retains sufficient expressivity; and (2) how to guarantee stable and effective optimization when updates are expressed in a random basis. To address the first challenge, we parameterize each weight update as $\Delta \boldsymbol{W} = \boldsymbol{L} \boldsymbol{Y} \boldsymbol{R}$, where $\boldsymbol{L} \in \mathbb{R}^{m \times a}$ and $\boldsymbol{R} \in \mathbb{R}^{b \times n}$ are fixed random projection matrices that induce a shared coordinate system across tasks. In this setup, the only trainable component is the compact core $\boldsymbol{Y} \in \mathbb{R}^{a \times b}$. This reduces the parameter count to $ab$, compared to $(m + n)r$ in LoRA-based methods, while preserving flexibility. For the second challenge, we analyze CoSA through the lens of the compressed sensing synthesis model $\boldsymbol{x} = \boldsymbol{\Psi} \boldsymbol{\alpha}$, where $\boldsymbol{\Psi} = \boldsymbol{R}^{\top} \otimes \boldsymbol{L}$ acts as the Kronecker dictionary (Broxson, 2006) and $\boldsymbol{\alpha} = \text{vec}(\boldsymbol{Y})$ are the parameters. As shown in Section 3, the induced dictionary satisfies the Restricted Isometry Property (RIP) under standard conditions, preserving the geometrical structure of the parameter space with a stable and well-conditioned optimization landscape. Our contributions are summarized as follows:

- We propose compressed sensing–based PEFT method with fixed random projections and a compact trainable core from a fundamentally different perspective compared to LoRA.

- A theoretical foundation is provided by framing CoSA as a synthesis process in compressed sensing, proving that its Kronecker dictionary of random projections satisfies the Restricted Isometry Property (RIP), ensuring near-isometry and stable optimization.

- Extensive experiments on NLU and NLG benchmarks with RoBERTa, LLaMA, and Qwen show that CoSA matches or outperforms state-of-the-art PEFT methods while offering substantial parameter savings.

## 2 RELATED WORK

Parameter-Efficient Fine-Tuning (PEFT) methods aim to adapt large pre-trained models to downstream tasks with minimal trainable parameters. LoRA (Hu et al., 2022) pioneered this approach by decomposing weight updates into low-rank matrices, achieving competitive performance to full fine-tuning with a significantly reduced number of trainable parameters. Building upon this foundation, AdaLoRA (Zhang et al., 2023) introduces adaptive rank allocation to dynamically adjust the importance of different parameter subsets during training. DoRA (Liu et al., 2024b) decomposes weights into magnitude and direction components, applying LoRA only to the directional component to better mimic full fine-tuning. Although this adaptation improves optimization stability, it retains the standard low-rank LoRA parameterization and does not alter the expressive structure of the update matrix.

While these methods often rely on random initialization of adapter modules, learning from random noise may lead to slow convergence. To address this issue, LoRA-GA (Wang et al., 2024) introduces an initialization method utilizing gradient approximation of the full weight matrix for faster convergence. Furthermore, PiSSA (Meng et al., 2024) proposes an alternative initialization strategy that leverages principal component analysis on pre-trained weights to accelerate convergence and also improve final performance.

Most LoRA-inspired methods restrict adaptation to a low-rank subspace through structural or adaptive parameterizations. In contrast, our approach leverages Compressed Sensing (CS) theory to represent weight updates in a fixed random basis with a compact trainable core, providing a stable and expressive alternative to low-rank parameterizations. Prior work has applied CS to compress gradients (Wang et al., 2018b; Li et al., 2020)for efficient training. However, CoSA is the first to formulate the weight update itself as a signal synthesized from a Kronecker-product random dictionary, leveraging the Restricted Isometry Property (RIP) to guarantee optimization stability in the compressed space.

## 3 PRELIMINARY

This section introduces the background of PEFT methods and core ideas of compressed sensing.

### 3.1 PARAMETER-EFFICIENT FINE-TUNING

We denote the full set of model parameters by $\Theta \in \mathbb{R}^D$. Generally, we have a base model with the full set of pre-trained parameters $\Theta_0 \in \mathbb{R}^D$. Full fine-tuning optimizes all parameters:

$$\Theta^* = \arg\min_{\Theta} \mathcal{L}(\Theta), \tag{1}$$

where $\mathcal{L}$ is the downstream task loss and $\Theta^*$ is the set of fully fine-tuned parameters.

The goal of PEFT is to match the performance of full fine-tuning while substantially reducing the number of trainable parameters. Unlike full fine-tuning, PEFT freezes the pre-trained parameters $\Theta_0$ and introduces a small set of trainable parameters $\Phi \in \mathbb{R}^d$ with $d \ll D$. These parameters specify a weight update through a mapping $g(\Phi)$, so that the adapted model becomes $\Theta = \Theta_0 + g(\Phi)$. The model adapts through a task-specific update $g(\Phi)$:

$$\Phi^* = \arg\min_{\Phi} \mathcal{L}(\Theta_0 + g(\Phi)). \tag{2}$$

For a particular weight matrix $\boldsymbol{W}_0 \in \mathbb{R}^{m \times n}$ within $\Theta_0$, we denote its update by $\Delta \boldsymbol{W} = g(\Phi)$ where $\Delta \boldsymbol{W} \in \mathbb{R}^{m \times n}$. Different PEFT methods differ in how $g(\Phi)$ (and thus $\Delta \boldsymbol{W}$) is defined. For example, LoRA (Hu et al., 2022) chooses $\Phi = \{\text{vec}(\boldsymbol{A}), \text{vec}(\boldsymbol{B})\}$ with $\boldsymbol{A} \in \mathbb{R}^{r \times n}$ and $\boldsymbol{B} \in \mathbb{R}^{m \times r}$, and defines $\Delta \boldsymbol{W} = \boldsymbol{B}\boldsymbol{A}$, a low-rank factorization with $\text{rank}(\Delta \boldsymbol{W}) \leq r \ll \min(m, n)$.

Thus, the key design question in PEFT is how to construct $g(\Phi)$ so that $\Delta \boldsymbol{W}$ is both compact and expressive. While LoRA and its variants define $g(\Phi)$ through low-rank factorization, this design choice may impose an inherent structural bottleneck that limits expressivity. They enforce that all task-specific adaptation must lie within an arbitrary (Hu et al., 2022) or selected (Meng et al., 2024) rank-$r$ subspace. When the essential optimization is dispersed across many distinct directions, such a constraint can create approximation errors and reduce expressivity. This structural bottleneck motivates alternative formulations that retain efficiency and provide greater expressivity. We aim to achieve this goal from a different view, utilizing the properties of compressed sensing. In Section 3.2, we discuss how we are inspired by exploring a different perspective based on compressed sensing.

### 3.2 COMPRESSED SENSING

Intrinsic dimensionality studies (Aghajanyan et al., 2020) show that fine-tuning relies on a surprisingly low-dimensional subspace of the full parameter space. A key property of such low-dimensional structures is that their geometry can be preserved under random projection into a moderately larger space, as formalized by the Johnson–Lindenstrauss lemma (Johnson et al., 1984). Compressed sensing (CS) generalizes this principle by showing that a high-dimensional signal that is sparse on some basis can be stably reconstructed from random linear projections that satisfy the *Restricted Isometry Property (RIP)* (Donoho, 2006; Candes & Tao, 2006; Candes et al., 2006; Candès et al., 2006). This connection motivates a compressed sensing-based formulation, where fixed random projections provide a universal dictionary to transfer the compressed signal.

Classical CS considers the problem of reconstructing a sparse signal $\boldsymbol{x} \in \mathbb{R}^p$ from a small set of linear measurements. Each measurement is a linear combination of the entries of $\boldsymbol{x}$, so collecting $m$ such measurements gives:

$$\boldsymbol{y} = \boldsymbol{\Phi}\boldsymbol{x}, \tag{3}$$

where $\boldsymbol{y} \in \mathbb{R}^m$ is observed low-dimensional measurements, $\boldsymbol{\Phi} \in \mathbb{R}^{m \times p}$ is the sensing matrix. Here, $p$ is the ambient dimension of the original signal, while $m$ is the number of measurements. $m \ll p$ holds because the number of measurements is limited in reality. Successful recovery is guaranteed when the measurement matrix $\boldsymbol{\Phi}$ satisfies the *Restricted Isometry Property (RIP)* (Candes & Tao, 2006; Candes et al., 2006), which ensures that reconstruction is stable without destroying the geometric structure of the initial parameter space.

**Restricted Isometry Property (RIP).** A measurement matrix $\mathbf{\Phi} \in \mathbb{R}^{m \times p}$ satisfies the RIP of order $s$ if there exists a constant $\delta_s \in (0, 1)$ such that for all $s$-sparse signals $\boldsymbol{x} \in \mathbb{R}^p$, following inequality holds:

$$(1 - \delta_s)\|\boldsymbol{x}\|_2^2 \ \leq \ \|\mathbf{\Phi}\boldsymbol{x}\|_2^2 \ \leq \ (1 + \delta_s)\|\boldsymbol{x}\|_2^2. \tag{4}$$

The smallest such $\delta_s$ is called the RIP constant of order $s$. Intuitively, RIP requires that $\mathbf{\Phi}$ approximately preserve the Euclidean norm of all $s$-sparse vectors, acting as a near-isometry on the set of sparse signals. This property ensures that distinct sparse vectors remain distinguishable after projection and that small perturbations in the coefficients translate into proportionally small changes in the projected signal. Consequently, RIP provides guarantees of structural preservation and stable recovery.

A central result in compressed sensing establishes that random matrices with entries sampled independently from Gaussian distributions satisfy RIP with high probability (Do et al., 2011; Zhang et al., 2018; Li et al., 2024). This theoretical guarantee justifies our choice of fixed random matrices as universal projection bases in CoSA, ensuring stable and reliable adaptation. Detailed explanations are listed in Appendix A.

While RIP is classically presented in the context of signal recovery, the same principle extends to the synthesis view (Elad, 2010; Bruckstein et al., 2009; Elad et al., 2007). In this perspective, a high-dimensional signal is generated as

$$\boldsymbol{x} = \mathbf{\Psi}\boldsymbol{\alpha}, \tag{5}$$

where $\mathbf{\Psi} \in \mathbb{R}^{p \times d}$ is a dictionary and $\boldsymbol{\alpha} \in \mathbb{R}^d$ the coefficient vector. If $\mathbf{\Psi}$ satisfies RIP, distinct sparse $\boldsymbol{\alpha}$ yield geometrically stable constructions of $\boldsymbol{x} \in \mathbb{R}^p$.

The recovery and synthesis views are mathematically equivalent under a change of basis, as detailed in Appendix A.1. This duality establishes random projections as principled tools for constructing expressive yet compact parameterizations. In Section 4, we leverage this perspective to design CoSA, which reinterprets PEFT updates through the lens of compressed sensing.

# 4 CoSA: Compressed Sensing-based Adaptation

This section introduces CoSA, combining compressed sensing with an efficient adapter design.

## 4.1 Overall Design

Traditional LoRA and its variants aim to train the update matrix $\Delta \boldsymbol{W} \in \mathbb{R}^{m \times n}$ as the low-rank representation $\Delta \boldsymbol{W} = \boldsymbol{B}\boldsymbol{A}$, where $\boldsymbol{A} \in \mathbb{R}^{r \times n}$ and $\boldsymbol{B} \in \mathbb{R}^{m \times r}$, as shown in Figure 1a. While $\boldsymbol{A}$ is initialized with a Gaussian or Kaiming random matrix, $\boldsymbol{B}$ matrix is initialized to zeros to ensure the update $\Delta \boldsymbol{W}$ starts from zeros. PiSSA assumes that principal singular values of pre-trained weight matrices can guide promising update directions of $\Delta \boldsymbol{W}$ and initializes $\boldsymbol{A}$ and $\boldsymbol{B}$ based on the singular value decomposition (SVD) of $\boldsymbol{W}_0$. Although the use of prior knowledge has proven effective both theoretically and empirically, it can potentially underperform in tasks where the initial model lacks suffcient knowledge. While our goal is not to explore the limitations of pioneer studies, we aim to propose a novel design that can perform and transfer well on broad and diverse tasks.

Our CoSA design originates from the classical compressed sensing technique. Inspired by the compression idea, we focus on a different perspective. The classical compressed sensing aims to reconstruct a target sparse signal $\boldsymbol{x}$ from a projection sensing matrix $\mathbf{\Phi}$ and a set of measurements $\boldsymbol{y}$. However, during the fine-tuning of a model, the *target* matrix is directly learned through the gradient descent process. Therefore, one can naturally think about utilizing the RIP to transfer the geometric structure of the target matrix into the measurement matrix as Equation 5. If we can obtain a low-dimensional $\boldsymbol{\alpha}$ through an optimization process, we can utilize a random projection *dictionary* $\mathbf{\Psi}$ to construct a *target* $\boldsymbol{x}$ while preserving the geometric structure. In the following paragraph, we discuss how this perspective maps to the PEFT world.

As depicted in Figure 1b, we denote the update weight matrix $\Delta \boldsymbol{W} \in \mathbb{R}^{m \times n}$ as the sparse *target matrix*, which can be represented as:

$$\Delta \boldsymbol{W} = \boldsymbol{L}\boldsymbol{Y}\boldsymbol{R}, \tag{6}$$

with fixed random projections $\boldsymbol{L} \in \mathbb{R}^{m \times a}$, $\boldsymbol{R} \in \mathbb{R}^{b \times n}$, and a trainable core $\boldsymbol{Y} \in \mathbb{R}^{a \times b}$. $\boldsymbol{L}$ and $\boldsymbol{R}$ are essential to maintain the RIP principal as the projection dictionary. Using a standard identity involving Kronecker product (Broxson, 2006) and vectorization operator, we can rewrite Equation 6 in vector form:

$$\text{vec}(\Delta \boldsymbol{W}) = (\boldsymbol{R}^\top \otimes \boldsymbol{L})\text{vec}(\boldsymbol{Y}), \tag{7}$$

where $\otimes$ denotes the Kronecker product and $\text{vec}(\cdot)$ is the vectorization operator. Let the vectorized weight update be $\boldsymbol{x} = \text{vec}(\Delta \boldsymbol{W}) \in \mathbb{R}^{mn}$, and the vectorized trainable core matrix be $\boldsymbol{\alpha} = \text{vec}(\boldsymbol{Y}) \in \mathbb{R}^{ab}$. Then the CoSA update process is equivalent to Equation 5: $\boldsymbol{x} = \boldsymbol{\Psi}\boldsymbol{\alpha}$, where the projection dictionary is given by $\boldsymbol{\Psi} = \boldsymbol{R}^\top \otimes \boldsymbol{L}$, and $\boldsymbol{\Psi} \in \mathbb{R}^{mn \times ab}$. Within this framework, the fine-tuning of CoSA is not a process of measuring a pre-existing $\Delta \boldsymbol{W}$, but rather one of learning the optimal coefficients $\boldsymbol{\alpha}$ (i.e., $\text{vec}(\boldsymbol{Y})$) within the low-dimensional subspace spanned by the dictionary $\boldsymbol{\Psi}$. The efficacy of this approach hinges on whether $\boldsymbol{\Psi}$ is a well-qualified dictionary, which means whether it satisfies the RIP. If two very different sparse matrices $\boldsymbol{\alpha}_1$ and $\boldsymbol{\alpha}_2$ generate nearly identical signals, i.e., $\boldsymbol{\Psi}\boldsymbol{\alpha}_1 \approx \boldsymbol{\Psi}\boldsymbol{\alpha}_2$, then the optimization landscape becomes ill-conditioned without RIP of $\boldsymbol{\Psi}$. Gradient-based optimization methods can be ineffective as any changes in $\boldsymbol{\alpha}$ may result in negligible changes in $\boldsymbol{x}$. If $\boldsymbol{\Psi}$ satisfies RIP, we will obtain the following equation:

$$\|\boldsymbol{\Psi}(\boldsymbol{\alpha}_1 - \boldsymbol{\alpha}_2)\|_2^2 \approx \|\boldsymbol{\alpha}_1 - \boldsymbol{\alpha}_2\|_2^2, \tag{8}$$

which guarantees stability during the fine-tuning process. In detail, small changes in $\boldsymbol{\alpha}$ yield proportionally small changes in $\boldsymbol{x}$, enabling stable and effective optimization.

**Theorem 1 (RIP of Kronecker Product Dictionaries)** *Let $\boldsymbol{\Psi}_1 \in \mathbb{R}^{a \times m}$ and $\boldsymbol{\Psi}_2 \in \mathbb{R}^{b \times n}$ be independent random matrices that satisfy the RIP for appropriate sparsity classes. Then their Kronecker product, $\boldsymbol{\Psi} = \boldsymbol{\Psi}_1^T \otimes \boldsymbol{\Psi}_2 \in \mathbb{R}^{mn \times ab}$, satisfies the RIP with high probability for the corresponding structured sparsity level.*

Theorem 1 provides the key theoretical justification for CoSA's design. A detailed proof can be found in (Duarte & Baraniuk, 2011). While Theorem 1 is a known result in Compressed Sensing, its application to PEFT provides the fundamental justification for CoSA's architectural design over existing random-basis methods. The RIP guarantee implies that the mapping from the trainable core $\boldsymbol{Y}$ to the weight update $\Delta \boldsymbol{W}$ is a near-isometry (Equation 7). This ensures that the optimization landscape is well-conditioned: distinct parameters in $\boldsymbol{Y}$ map to distinct updates in $\Delta \boldsymbol{W}$, and gradients propagate without vanishing or exploding. This protects CoSA from the degeneracy issues often faced by methods that learn the basis from scratch. Crucially, this stability allows CoSA to train a dense core matrix $\boldsymbol{Y}$ rather than simple scaling vectors as VeRA (Kopiczko et al., 2023). Theorem 1 guarantees that the complex linear combinations of basis vectors formed by $\boldsymbol{Y}$ remain distinguishable and robust. This enables *subspace mixing*, where input features from $\boldsymbol{R}$ can be routed to any output direction in $\boldsymbol{L}$. Thus, it provides significantly higher expressivity and full-rank behavior within the compressed subspace compared to the diagonal *subspace scalings* of vector-based methods.

We now formalize the CoSA framework within this perspective. CoSA introduces an additive, compressed adaptation module to the linear layers of a model designated for fine-tuning. Let $\boldsymbol{X} \in \mathbb{R}^n$ and $\boldsymbol{Z} \in \mathbb{R}^m$ be the input and output of a standard linear layer, the forward computation of a base model is $\boldsymbol{Z} = \boldsymbol{W}_0\boldsymbol{X}$, where $\boldsymbol{W}_0 \in \mathbb{R}^{m \times n}$ is the pre-trained weight matrix. With CoSA, the forward pass becomes:

$$\boldsymbol{Z} = \boldsymbol{W}_0\boldsymbol{X} + \boldsymbol{L}(\boldsymbol{Y}(\boldsymbol{R}\boldsymbol{X})), \tag{9}$$

with $\boldsymbol{L} \in \mathbb{R}^{m \times a}$ and $\boldsymbol{R} \in \mathbb{R}^{b \times n}$ being frozen, and $\boldsymbol{Y} \in \mathbb{R}^{a \times b}$ being trainable. $\boldsymbol{Y}$ is initialized as zeros to ensure that the model initially behaves as the pre-trained model. $\boldsymbol{L}$ and $\boldsymbol{R}$ are initialized with Gaussian random matrices to satisfy RIP as discussed above. This forward process consists of three stages:

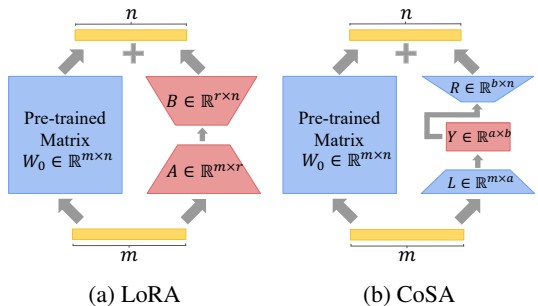

(a) LoRA        (b) CoSA

Figure 1: Comparison of LoRA and CoSA. LoRA constrains updates to a low-rank subspace via matrices $\boldsymbol{A}$ and $\boldsymbol{B}$, while CoSA reinterprets updates as a compressed sensing process with fixed projections $\boldsymbol{L}$, $\boldsymbol{R}$ and a compact trainable core $\boldsymbol{Y}$.

Table 1: Comparison of trainable parameters and training complexities across different methods.

| Method | Trainable parameters | Optimizer state | Forward/Backward | Storage |
|---|---|---|---|---|
| LoRA $(r)$ | $(m+n)r$ | $\mathcal{O}((m+n)r)$ | $\mathcal{O}(mn)$ | $\mathcal{O}((m+n)r)$ |
| PiSSA $(r)$ | $(m+n)r$ | $\mathcal{O}((m+n)r)$ | $\mathcal{O}(mn)$ | $\mathcal{O}((m+n)r)$ |
| CoSA $(a,b)$ | $ab$ | $\mathcal{O}(ab)$ | $\mathcal{O}(mn)$ | $\mathcal{O}(ab)$ |

1. Input Compression: $\boldsymbol{u} = \boldsymbol{R}\boldsymbol{X} \in \mathbb{R}^b$. The input $\boldsymbol{X}$ is projected by the fixed matrix $\boldsymbol{R}$ into a low-dimensional space with the initial output $\boldsymbol{u}$.

2. Core Transformation: $\boldsymbol{v} = \boldsymbol{Y}\boldsymbol{u} \in \mathbb{R}^a$. The trainable core matrix $\boldsymbol{Y}$ performs a learned transformation into another low-dimensional space with the intermediate result $\boldsymbol{v}$.

3. Output Reconstruction: $\Delta\boldsymbol{W}\boldsymbol{X} = \boldsymbol{L}\boldsymbol{v} \in \mathbb{R}^m$. The fixed matrix $\boldsymbol{L}$ projects the intermediate result $\boldsymbol{v}$ back into the original output space, yielding the final reconstruction.

Freezing $\boldsymbol{L}$ and $\boldsymbol{R}$ establishes a task-agnostic, shared coordinate system in which all tasks express their updates via the same dictionary $\boldsymbol{R}^\top \otimes \boldsymbol{L}$; only the small core $\boldsymbol{Y}$ changes across tasks. This decouples *where* adaptation lives (the basis) from *what* is adapted (the coefficients), enabling plug-and-play reuse and warm-starts of $\boldsymbol{Y}$ between tasks. RIP ensures that $\boldsymbol{L}$ and $\boldsymbol{R}$ projections provide stability and prevent degenerate optimization landscapes. This favors transferability of CoSA: the same random projections support many tasks with only $\boldsymbol{Y}$ retrained.

As $\boldsymbol{W}_0$, $\boldsymbol{L}$, and $\boldsymbol{R}$ are frozen and the backpropagation algorithm only needs to compute the gradient for the loss $\ell$ with respect to the trainable matrix $\boldsymbol{Y}$. Let $\boldsymbol{g} = \partial\mathcal{L}/\partial\boldsymbol{Z} \in \mathbb{R}^m$. By the chain rule, we derive the gradient for $\boldsymbol{Y}$:

$$\frac{\partial\mathcal{L}}{\partial\boldsymbol{Y}} = (\boldsymbol{L}^\top\boldsymbol{g})(\boldsymbol{R}\boldsymbol{X})^\top, \tag{10}$$

The gradient $\nabla\boldsymbol{Y}$ is the outer product of an $a$-dimensional vector $(\boldsymbol{L}^\top\boldsymbol{g})$ and a $b$-dimensional vector $(\boldsymbol{R}\boldsymbol{X})^\top$. In practice, $a$ and $b$ are smaller than the input and output dimensions of the linear layer that $a < m$ and $b < n$, ensuring the parameter-efficiency training of CoSA.

After training, only the compact matrix $\boldsymbol{Y}$ needs to be stored as the adapter module, together with a random seed for regenerating $\boldsymbol{L}$ and $\boldsymbol{R}$ during inference. This design avoids storing large projection matrices explicitly, keeping the storage footprint minimal while ensuring reproducibility of the random basis.

### 4.2 PARAMETER EFFICIENCY

One of the main advantages in PEFT world is the reduced number of trainable parameters. LoRA and PiSSA scale their parameter counts linearly with the input and output dimensions of a layer, requiring $(m+n)r$ additional parameters. For large attention projections and fully connected layers, this results in substantial overhead even with modest ranks. In contrast, CoSA decouples the parameter count from $m$ and $n$. The number of trainable parameters depends only on the projection dimensions (i.e., $ab$). $a$ and $b$ are manually specified compression dimensions, typically chosen such that $r \le a < m$ and $r \le b < n$. Table 1 summarizes the parameter and complexity characteristics of each method.

All three methods introduce additional matrix multiplications per layer, so the asymptotic forward and backward complexities remain $\mathcal{O}(mn)$, dominated by the multiplication with the frozen weight $\boldsymbol{W}_0$. The key distinction emerges in the optimizer state and memory usage of the trainable parameters. Since optimizer states scale linearly with the number of trainable parameters, LoRA and PiSSA require $(m+n)r$ parameters per layer and around $3(m+n)r$ states for optimizers such as Adam (Kingma, 2014) or AdamW (Loshchilov & Hutter, 2017). By contrast, CoSA maintains only $ab$ parameters and around $3ab$ optimizer states that are completely independent of the input and output dimensions $(m, n)$. This makes CoSA more memory-efficient when applied to wide layers in large-scale language models, while retaining the same computational complexity as low-rank methods. We provide an empirical analysis of the memory cost of the adaptation modules in Section 5.3.2 for further verification. Moreover, the fixed projection matrices $\boldsymbol{L}$ and $\boldsymbol{R}$ do not need to be stored explicitly. Instead, they can be generated on demand from a saved random seed, further reducing

storage overhead and simplifying deployment. As a result, CoSA adapters are lightweight, portable, and easily integrated into different models or tasks without additional overhead.

## 5 EVALUATION

We conduct comprehensive experiments to evaluate CoSA and state-of-the-art PEFT methods across diverse tasks and model architectures in this section.

### 5.1 EXPERIMENTAL SETUP

**Baselines, Models and Benchmarks.** We compare CoSA against four representative approaches: *Full fine-tuning*, *LoRA* (Hu et al., 2022), *AdaLoRA* (Zhang et al., 2023) and *PiSSA* (Meng et al., 2024). We follow PiSSA's experimental setup with minor adjustments to accommodate differences in models and hardware. Details are provided in Appendix C.

For Natural Language Understanding (NLU), we evaluate CoSA on the GLUE benchmark (Wang et al., 2018a) using RoBERTa$_{base}$ and RoBERTa$_{large}$ (Liu et al., 2019), covering SST-2, MRPC, CoLA, QNLI, RTE, and STS-B tasks. For Natural Language Generation (NLG), we experiment with Llama-3.2-1B, Llama-3.1-8B (Grattafiori et al., 2024), and Qwen2-7B (Yang et al., 2024a). To assess mathematical reasoning and code generation abilities, these models are fine-tuned on two instruction datasets: MetaMathQA (Yu et al., 2023), Code-Feedback (Zheng et al., 2024). We use the 100K subsets for all NLG datasets following PiSSA's setup. Evaluation is performed on GSM8K (Cobbe et al., 2021) and MATH (Hendrycks et al., 2021) for reasoning, and HumanEval (Chen et al., 2021) and MBPP (Austin et al., 2021) for code generation.

**Evaluation Metrics.** We adopt standard metrics for each benchmark category. For mathematical reasoning, we report accuracy on both GSM8K and MATH, which measures exact match accuracy over formal mathematical problem solving. For code generation, we report Pass@1, the proportion of top-1 generated programs that pass all test cases, on HumanEval and MBPP. For natural language understanding, we follow the official GLUE evaluation protocol (Wang et al., 2018a). Specifically, we report Matthews correlation for CoLA, F1 score for MRPC, the average of Pearson and Spearman correlations for STS-B, and accuracy for all other tasks. All results are averaged over 3 runs with different random seeds and reported with the standard deviations.

### 5.2 EXPERIMENTAL RESULTS

**Natural Language Understanding.** Table 2 presents results on the GLUE benchmark with RoBERTa$_{base}$ and RoBERTa$_{large}$. Across nearly all tasks, CoSA consistently outperforms other PEFT baselines. On MRPC, CoSA improves over the strongest baseline by 0.36 with RoBERTa$_{base}$ and by 0.22 with RoBERTa$_{large}$ on the F1 score. Overall, CoSA achieves best or second-best performance across a wide range of NLU tasks. Importantly, these improvements are consistent across model scales, indicating the method's scalability.

**Natural Language Generation.** Table 3 reports results on mathematical reasoning and code generation benchmarks across multiple model scales. CoSA consistently outperforms LoRA and AdaLoRA, and in most cases matches or exceeds PiSSA, without task-specific initialization or pre-processing. For example, on LLaMA-3.2-1B, CoSA improves average performance from 27.75 with PiSSA to 28.10, producing a more stable accuracy across all four tasks. On the larger LLaMA-3.1-8B, CoSA achieves an average of 56.11, closely matching PiSSA while outperforming LoRA by 3.0% and AdaLoRA by 6.8%. Furthermore, on Qwen2-7B, CoSA demonstrates the highest overall score of 66.83, with consistent advantages on GSM8K and HumanEval. These results indicate that compressed representations in a fixed random basis are capable of capturing task-specific adaptations more effectively and reliably than low-rank decompositions. Moreover, the relative strength of CoSA is still pronounced as the model scale increases. Performance remains competitive or superior on both LLaMA-3.1-8B and Qwen2-7B. These results confirm that CoSA serves as a robust and scalable adaptation method, providing state-of-the-art performance in natural language generation while maintaining parameter efficiency. More results are listed in Appendix E.

Table 2: Performance comparison on NLU tasks in the GLUE benchmark. Results show accuracy (%) for classification tasks, Pearson correlation for STS-B, and Matthews correlation for CoLA.

| Method | # Trainable Params | SST-2 | MRPC | CoLA | QNLI | RTE | STS-B | Avg |
|---|---|---|---|---|---|---|---|---|
| | | | | $RoBERTa_{base}$ | | | | |
| Full FT | 125M | $93.69_{\pm 0.12}$ | $86.39_{\pm 1.14}$ | $46.32_{\pm 0.93}$ | $92.26_{\pm 0.21}$ | $69.91_{\pm 1.05}$ | $86.66_{\pm 0.18}$ | 79.21 |
| LoRA | 1.03M | $93.73_{\pm 0.46}$ | $88.33_{\pm 0.30}$ | $53.95_{\pm 0.88}$ | $89.99_{\pm 0.71}$ | $72.80_{\pm 1.63}$ | $89.69_{\pm 0.17}$ | 81.42 |
| AdaLoRA | 1.26M | $93.46_{\pm 0.11}$ | $89.75_{\pm 0.47}$ | $54.63_{\pm 0.76}$ | $88.63_{\pm 0.91}$ | $73.29_{\pm 1.66}$ | $90.72_{\pm 0.04}$ | 81.75 |
| PiSSA | 1.03M | $93.27_{\pm 0.69}$ | $89.56_{\pm 0.52}$ | $57.39_{\pm 1.34}$ | $88.57_{\pm 1.02}$ | $73.11_{\pm 3.32}$ | $89.60_{\pm 0.18}$ | 81.92 |
| VeRA | 0.75M | $\mathbf{94.00}_{\pm 0.19}$ | $90.98_{\pm 0.52}$ | $\mathbf{60.67}_{\pm 0.80}$ | $92.25_{\pm 0.24}$ | $72.56_{\pm 2.52}$ | $\mathbf{90.48}_{\pm 0.15}$ | $\mathbf{83.50}$ |
| DoRA | 3.35M | $92.47_{\pm 0.76}$ | $89.89_{\pm 0.42}$ | $48.59_{\pm 2.70}$ | $88.45_{\pm 0.35}$ | $\mathbf{76.77}_{\pm 1.63}$ | $90.46_{\pm 0.17}$ | 81.11 |
| **CoSA** | 1.18M | $93.12_{\pm 0.40}$ | $\mathbf{91.34}_{\pm 0.50}$ | $58.79_{\pm 0.76}$ | $91.09_{\pm 0.50}$ | $74.85_{\pm 2.71}$ | $90.21_{\pm 0.10}$ | 83.23 |
| | | | | $RoBERTa_{large}$ | | | | |
| Full FT | 355M | $95.42_{\pm 0.07}$ | $85.41_{\pm 0.22}$ | $56.26_{\pm 2.10}$ | $94.28_{\pm 0.28}$ | $80.51_{\pm 1.44}$ | $87.35_{\pm 1.83}$ | 83.21 |
| LoRA | 8.16M | $96.06_{\pm 0.24}$ | $90.42_{\pm 0.38}$ | $\mathbf{65.29}_{\pm 1.07}$ | $\mathbf{94.62}_{\pm 0.28}$ | $76.17_{\pm 0.82}$ | $90.44_{\pm 0.13}$ | 85.50 |
| AdaLoRA | 8.16M | $\mathbf{96.10}_{\pm 0.23}$ | $92.36_{\pm 0.19}$ | $59.07_{\pm 1.25}$ | $91.87_{\pm 0.41}$ | $\mathbf{85.32}_{\pm 0.54}$ | $91.70_{\pm 0.09}$ | 86.07 |
| PiSSA | 8.16M | $95.37_{\pm 0.18}$ | $91.53_{\pm 0.81}$ | $58.61_{\pm 1.27}$ | $93.30_{\pm 0.29}$ | $81.47_{\pm 0.55}$ | $90.69_{\pm 0.30}$ | 85.16 |
| VeRA | 1.31M | $94.80_{\pm 0.28}$ | $81.22_{\pm 0.01}$ | $64.25_{\pm 1.12}$ | $92.51_{\pm 1.89}$ | $82.31_{\pm 1.44}$ | $89.58_{\pm 0.84}$ | 84.11 |
| DoRA | 8.38M | $94.44_{\pm 0.28}$ | $91.88_{\pm 0.43}$ | $62.71_{\pm 0.70}$ | $92.23_{\pm 0.03}$ | $84.48_{\pm 0.36}$ | $\mathbf{92.24}_{\pm 0.06}$ | 86.23 |
| **CoSA** | 6.19M | $95.11_{\pm 0.58}$ | $\mathbf{92.48}_{\pm 0.51}$ | $63.07_{\pm 0.66}$ | $93.78_{\pm 0.47}$ | $84.60_{\pm 1.46}$ | $91.88_{\pm 0.18}$ | $\mathbf{86.82}$ |

Table 3: Performance comparison across different model scales. Results show accuracy (%) for GSM8K and MATH, and pass@1 (%) for HumanEval and MBPP.

| Model | Method | # Trainable Params | GSM8K | MATH | HumanEval | MBPP | Average |
|---|---|---|---|---|---|---|---|
| | Full FT | 1,236M | $44.95_{\pm 0.26}$ | $9.83_{\pm 0.63}$ | $29.27_{\pm 1.64}$ | $38.37_{\pm 0.97}$ | 30.61 |
| LLaMA-3.2-1B | LoRA | 90M | $30.59_{\pm 0.43}$ | $5.84_{\pm 0.12}$ | $21.53_{\pm 0.97}$ | $37.67_{\pm 1.50}$ | 23.91 |
| | AdaLoRA | 113M | $33.10_{\pm 0.98}$ | $6.59_{\pm 0.37}$ | $21.77_{\pm 1.56}$ | $36.57_{\pm 1.02}$ | 24.51 |
| | PiSSA | 90M | $37.85_{\pm 0.32}$ | $\mathbf{7.83}_{\pm 0.43}$ | $26.40_{\pm 1.39}$ | $\mathbf{38.90}_{\pm 0.70}$ | 27.75 |
| | **CoSA** | 29M | $\mathbf{39.45}_{\pm 0.99}$ | $7.83_{\pm 0.40}$ | $\mathbf{26.83}_{\pm 3.65}$ | $38.27_{\pm 3.25}$ | $\mathbf{28.10}$ |
| | Full FT | 8,030M | $76.47_{\pm 1.17}$ | $25.21_{\pm 0.40}$ | $56.87_{\pm 3.36}$ | $59.53_{\pm 1.46}$ | 54.52 |
| LLaMA-3.1-8B | LoRA | 336M | $72.55_{\pm 0.72}$ | $25.57_{\pm 0.22}$ | $51.20_{\pm 0.60}$ | $\mathbf{68.63}_{\pm 0.29}$ | 54.49 |
| | AdaLoRA | 419M | $69.25_{\pm 0.78}$ | $23.81_{\pm 0.13}$ | $49.00_{\pm 0.75}$ | $68.07_{\pm 1.29}$ | 52.53 |
| | PiSSA | 336M | $77.03_{\pm 0.86}$ | $\mathbf{27.60}_{\pm 0.55}$ | $54.67_{\pm 4.72}$ | $66.20_{\pm 1.56}$ | $\mathbf{56.38}$ |
| | **CoSA** | 58M | $\mathbf{77.18}_{\pm 2.27}$ | $26.99_{\pm 0.76}$ | $55.07_{\pm 3.60}$ | $65.20_{\pm 0.66}$ | 56.11 |
| Qwen2-7B | Full FT | 7,615M | $76.93_{\pm 0.19}$ | $33.07_{\pm 0.47}$ | $69.93_{\pm 2.75}$ | $69.33_{\pm 1.67}$ | 62.32 |
| | LoRA | 323M | $81.45_{\pm 0.47}$ | $\mathbf{45.67}_{\pm 0.16}$ | $70.10_{\pm 0.60}$ | $69.67_{\pm 0.81}$ | 66.72 |
| | AdaLoRA | 404M | $78.80_{\pm 0.54}$ | $44.39_{\pm 0.29}$ | $\mathbf{71.33}_{\pm 1.85}$ | $70.10_{\pm 0.97}$ | 66.16 |
| | PiSSA | 323M | $81.37_{\pm 0.32}$ | $44.62_{\pm 0.20}$ | $70.33_{\pm 1.99}$ | $\mathbf{70.47}_{\pm 2.91}$ | 66.70 |
| | **CoSA** | 51M | $\mathbf{81.50}_{\pm 1.07}$ | $45.09_{\pm 0.15}$ | $71.33_{\pm 0.65}$ | $69.40_{\pm 2.13}$ | $\mathbf{66.83}$ |

## 5.3 ABLATION STUDY

We conduct a comprehensive ablation study of compression parameters and the memory cost of the adaptation modules. The theoretical effectiveness of RIP is provided in Appendix A.

### 5.3.1 STUDY OF COMPRESSION PARAMETERS $a$ AND $b$

A central design choice in CoSA is the selection of compression dimensions $(a, b)$, which define the size of the trainable core $Y \in \mathbb{R}^{a \times b}$. Since the number of trainable parameters scales as $ab$, these dimensions directly control the expressivity. To study this effect, we conduct a systematic study by varying $a$ and $b$ across a broad range while fixing the Llama-3.2-1B model and the same training setup. Figure 2 reports the results as a heatmap, where each cell corresponds to the average performance on GSM8K and MATH. The performance increases rapidly as $(a, b)$ grow from very small values. For example, raising from (32,32) to (128,128) boosts average accuracy from 10.3% to 18.1%, nearly doubling performance. However, the improvements plateau once $(a, b)$ become sufficiently large. Increasing from (1024,1024) at 25.8% to (2048,2048) at 25.6% yields a slight decline despite a fourfold increase in parameters.

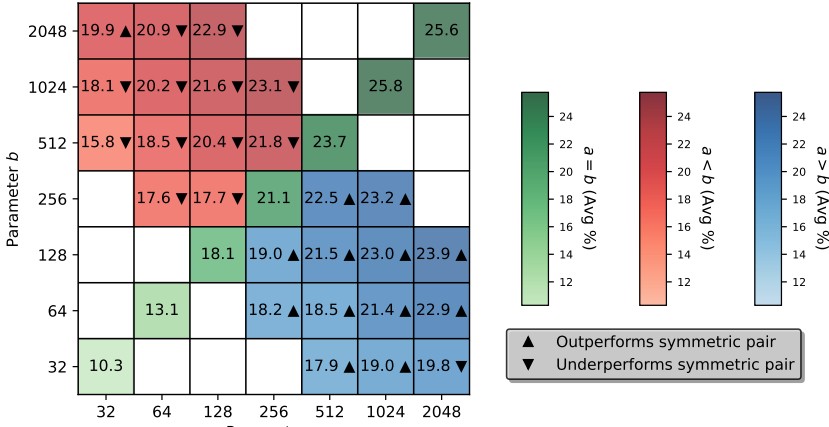

Figure 2: Performance across compression pairs $(a, b)$. Blue: $a > b$, red: $a < b$, green diagonal: $a = b$. ▲/▼ mark configurations that outperform/underperform their symmetric counterparts. Color intensity reflects score magnitude.

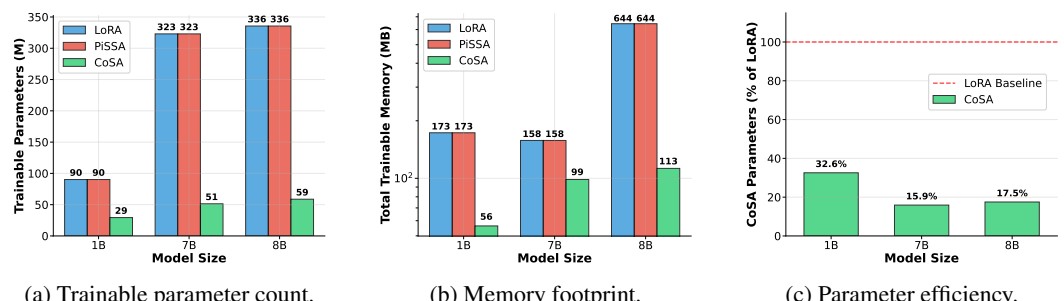

(a) Trainable parameter count.  (b) Memory footprint.  (c) Parameter efficiency.

Figure 3: Comparison of parameter and memory efficiency across LoRA, PiSSA, and CoSA. (a) Trainable parameter count across model sizes, showing CoSA requires fewer parameters. (b) Memory footprint of adaptation modules (including optimizer states), where CoSA achieves substantial savings. (c) Relative efficiency of CoSA expressed as a percentage of LoRA's trainable parameters, highlighting reductions across models (*1B: Llama-3.2-1B*; *7B: Qwen2-7B*; *8B: Llama-3.1-8B*).

The heatmap further highlights asymmetry between $a$ and $b$. Symmetric comparisons (e.g., (512,128) vs. (128,512)) show that enlarging $a$ yields more consistent benefits: (512,128) outperforms (128,512) by 5.4%. This effect aligns with the role of $a$, which controls the input projection and determines how richly the incoming feature space is represented after random projection $\boldsymbol{R}$. In conclusion, these results demonstrate that CoSA remains effective across diverse compression configurations and that enlarging the input-side dimension tends to provide more consistent benefits than enlarging the output-side dimension.

### 5.3.2 PARAMETER EFFICIENCY

Figure 3 compares the parameter counts and memory costs by the adaptation modules of LoRA, PiSSA, and CoSA across different model scales during training on the MetaMath dataset. The training is conducted under the same configurations as the main experiments in Table 3. We employ the rank of 128 for LoRA and PiSSA and $(a, b)$ of (1024,256) for CoSA to ensure fairness in expressivity. CoSA shows a consistent reduction in the number of trainable parameters. For Qwen2-7B, CoSA requires only 51M parameters compared to 323M for LoRA and PiSSA. This reduction directly translates into smaller memory overheads. At the 8B scale, the adaptation modules of LoRA and PiSSA consume 644MB of memory (including optimizer states), whereas CoSA reduces this to 236MB, cutting the memory footprint by more than 60%. Compared to LoRA, CoSA operates with less than 32.6% of the parameters across all employed models.

## 6   CONCLUSION

In this work, we introduced CoSA, a compressed sensing–based approach to parameter-efficient fine-tuning that replaces learned low-rank factors with fixed random projections and a compact trainable core. By bridging compressed sensing with PEFT, CoSA provides a principled design that preserves expressivity while substantially reducing parameter and optimizer state requirements. Our theoretical analysis shows that the induced Kronecker dictionary satisfies the Restricted Isometry Property (RIP), ensuring stable and geometrically meaningful optimization. Extensive experiments across natural language understanding and generation benchmarks demonstrate that CoSA achieves competitive or superior performance compared to state-of-the-art baselines, confirming its effectiveness as a practical and theoretically grounded method for adapting large language models.

## 7   LIMITATION

Despite strong benchmark performance, CoSA has two main limitations. First, its effectiveness depends on compression parameters $(a, b)$, whose optimal values are task-dependent and may require extensive tuning despite our ablation guidance. Second, our evaluation is limited to NLU, math reasoning, and code generation. Extending CoSA to a wider range, including evaluations on visual, cross-lingual, and multimodal benchmarks, remains important future work.

ETHICS STATEMENT

In this paper, we propose CoSA, a compressed sensing–based method for parameter-efficient fine-tuning of large language models (LLMs). Our approach achieves promising performance while enabling efficient adaptation of LLMs. By reducing the number of trainable parameters, CoSA promotes the accessibility and democratization of advanced language technologies, particularly for researchers and practitioners with limited computational resources. This improvement contributes positively to the sustainability of machine learning by reducing energy consumption during training and deployment.

Although we do not anticipate any immediate negative ethical implications from our approach, it's important to acknowledge that machine learning technologies, including LLMs, have broader impacts. The increased accessibility of fine-tuned models underscores the need for ongoing research into potential biases inherited from pre-trained models and the development of robust safeguards. Our method focuses on performance and parameter efficiency rather than addressing underlying biases in LLMs. We encourage users to conduct appropriate evaluations before deployment.

All authors have carefully reviewed and adhered to the ICLR Code of Ethics. We affirm that our study complies with research integrity standards and raises no issues regarding human subjects, privacy, or legal compliance. We encourage future work to explore complementary safeguards that align with the responsible use of increasingly efficient and accessible LLM technologies.

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

## A THEORETICAL FOUNDATIONS OF CoSA

This appendix provides detailed mathematical derivations and analysis supporting the theoretical claims in the main paper. We present the rigorous foundations underlying CoSA's compressed sensing framework, including two views of compressed sensing, RIP bound derivations, empirical measurement methodologies, and mathematical structure analysis.

## A.1 RECOVERY VS. SYNTHESIS: WHY ARE THEY EQUIVALENT?

The classical view of compressed sensing is the *recovery model*, in which the goal is to reconstruct a high-dimensional but sparse signal from a small number of linear measurements. Opposite to this is the *synthesis model*, where the signal itself is generated from a sparse set of coefficients in a fixed dictionary. Although framed differently, the two views are mathematically equivalent under a change of basis.

**Recovery model.** In the standard formulation, we observe:

$$y = \Phi x, \tag{11}$$

where $x \in \mathbb{R}^p$ is the high-dimensional signal, $y \in \mathbb{R}^m$ are the $m$ observed measurements, and $\Phi \in \mathbb{R}^{m \times p}$ is the sensing matrix with $m \ll p$. The assumption is that $x$ is $s$-sparse in the canonical basis (i.e., the identity matrix).

**Synthesis model.** Suppose instead that $x$ is not sparse in the canonical basis but is sparse in some dictionary $\Psi \in \mathbb{R}^{p \times d}$. Then $x$ can be expressed as:

$$x = \Psi \alpha, \quad \alpha \in \mathbb{R}^d \text{ is } s\text{-sparse.} \tag{12}$$

Substituting into the measurement equation gives:

$$y = \Phi x = \Phi \Psi \alpha = A \alpha, \qquad A = \Phi \Psi \in \mathbb{R}^{m \times d}. \tag{13}$$

Thus, recovering a signal $x$ that is sparse in the basis $\Psi$ is equivalent to recovering sparse coefficients $\alpha$ under the effective sensing matrix $A$:

Recovery problem for $x$ sparse in $\Psi$ $\Leftrightarrow$ Synthesis problem with $x = \Psi \alpha$ and $\alpha$ sparse.

The sensing matrix $\Phi$ in the recovery model and the dictionary $\Psi$ in the synthesis model both serve as mappings between low-dimensional and high-dimensional spaces. This duality establishes that recovery and synthesis are mathematically interchangeable, providing the theoretical justification for adopting the synthesis perspective in our CoSA framework.

## A.2 DERIVATION OF THEORETICAL RIP BOUNDS

We derive the fundamental theoretical estimate:

$$\delta_s \leq C\sqrt{\frac{s \log(n)}{m}},$$

which underpins CoSA's stability guarantees. This bound is classical in compressed sensing (Candes & Tao, 2006; Candès et al., 2006; Baraniuk et al., 2010) and shows that random projections preserve the structure of all $s$-sparse vectors with high probability, provided the number of measurements $m$ is large enough.

**Problem Setup.** Let $\Phi \in \mathbb{R}^{m \times n}$ be a Gaussian random matrix with entries $\Phi_{ij} \sim \mathcal{N}(0, 1/n)$. The Restricted Isometry Property (RIP) of order $s$ requires that for every $s$-sparse vector $\alpha$:

$$(1 - \delta_s)\|\alpha\|_2^2 \leq \|\Phi \alpha\|_2^2 \leq (1 + \delta_s)\|\alpha\|_2^2. \tag{14}$$

Here, $\delta_s$ is the smallest RIP constant for which Equation 14 holds, and it measures how close $\Phi$ is to an isometry on the set of all $s$-sparse vectors.

For a *fixed* $s$-sparse unit vector $\alpha$, the random variable $\|\Phi \alpha\|_2^2$ is the average of $m$ independent $\chi^2$-like variables. Its expectation is exactly $\|\alpha\|_2^2 = 1$. Classical concentration inequalities (Hoeffding, 1963) guarantee that:

$$\Pr\left(\left|\|\Phi \alpha\|_2^2 - 1\right| \geq t\right) \leq 2 \exp\left(-\frac{mt^2}{C_1}\right), \tag{15}$$

for some universal constant $C_1 > 0$. Intuitively, this means that with probability exponentially close to 1, the distortion of a single vector is at most $t$.

RIP requires Equation 14 to hold *simultaneously for all* $s$-sparse vectors, not just one. The set of $s$-sparse unit vectors is infinite, so we discretize it using an $\epsilon$-net. An $\epsilon$-net is a finite subset of vectors such that every $s$-sparse unit vector lies within $\epsilon$ distance of some net vector.

**Lemma 1 (Sparse Vector Covering (Vershynin, 2018))** *The set of $s$-sparse unit vectors in $\mathbb{R}^n$ can be covered by an $\epsilon$-net of size at most:*

$$\mathcal{N}(\epsilon) \leq \binom{n}{s} \left(\tfrac{3}{\epsilon}\right)^s. \tag{16}$$

Using the approximation $\binom{n}{s} \leq (en/s)^s$ and setting $\epsilon = 1/2$ gives:

$$\mathcal{N}(1/2) \leq (6en/s)^s.$$

We now apply the union bound over all vectors in the $\epsilon$-net. For each vector, inequality Equation 15 holds with high probability. The union bound ensures it holds simultaneously for all vectors in the net:

$$\Pr(\delta_s \geq t) \ \leq \ \mathcal{N}(1/2) \cdot 2 \exp\left(-\frac{mt^2}{C_1}\right) + \text{approximation error}.$$

The additional approximation error accounts for shifting from the finite net back to the entire set of sparse vectors. It is proportional to $\frac{1}{2}t$ due to the net's granularity.

Substituting $\mathcal{N}(1/2)$, we obtain:

$$\Pr(\delta_s \geq t) \ \leq \ (6en/s)^s \cdot 2 \exp\left(-\frac{mt^2}{C_1}\right) + \frac{1}{2}t.$$

To make the failure probability small (say, less than $\eta = 0.01$), we require:

$$t \gtrsim C\sqrt{\frac{s\log(n)}{m}}.$$

Thus,

$$\delta_s \ \leq \ C\sqrt{\frac{s\log(n)}{m}}, \tag{17}$$

with probability at least $1 - \eta$, where $C$ is an absolute constant depending only on $C_1$ and $\eta$.

**Implications for CoSA.** In our setting, CoSA uses a Kronecker dictionary $\boldsymbol{\Psi} = \boldsymbol{R}^\top \otimes \boldsymbol{L}$. The same RIP analysis applies by interpreting:

- $m$ = effective number of measurements (degrees of freedom from the Kronecker projections),
- $n = ab$ = ambient dimension of the coefficient space $\text{vec}(\boldsymbol{Y})$,
- $s$ = effective sparsity level of the representation.

This theoretical bound justifies that CoSA's random projection design inherits RIP guarantees, ensuring that optimization remains stable and expressive.

### A.3 EMPIRICAL RIP MEASUREMENT METHODOLOGY

We present the detailed methodology for empirically measuring RIP constants and establish its theoretical justification.

#### A.3.1 MONTE CARLO ESTIMATOR DESIGN

Given a specific matrix realization $\boldsymbol{\Phi}$, the true RIP constant is:

$$\delta_s^{\text{true}} = \max_{\boldsymbol{\alpha}:\|\boldsymbol{\alpha}\|_0=s} \left| \frac{\|\boldsymbol{\Phi}\boldsymbol{\alpha}\|_2^2}{\|\boldsymbol{\alpha}\|_2^2} - 1 \right| \tag{18}$$

Since this optimization is computationally intractable, we approximate it via sampling.

We generate $N$ independent $s$-sparse vectors $\{\boldsymbol{\alpha}_1, \boldsymbol{\alpha}_2, \ldots, \boldsymbol{\alpha}_N\}$ according to:

---

**Algorithm 1** Sparse Vector Generation

---

1: **for** $i = 1$ to $N$ **do**
2:    $\boldsymbol{\alpha}_i \leftarrow \mathbf{0} \in \mathbb{R}^n$
3:    $\mathcal{S}_i \leftarrow \text{UniformRandomSubset}(\{1, \ldots, n\}, s)$
4:    **for** $j \in \mathcal{S}_i$ **do**
5:       $(\boldsymbol{\alpha}_i)_j \leftarrow \mathcal{N}(0, 1)$
6:    **end for**
7:    $r_i \leftarrow \|\boldsymbol{\Phi}\boldsymbol{\alpha}_i\|_2^2 / \|\boldsymbol{\alpha}_i\|_2^2$
8: **end for**
9: **return** $\{r_1, r_2, \ldots, r_N\}$

---

The empirical RIP constant is computed as (Tucker, 1959):

$$\delta_s^{\text{empirical}} = \text{percentile}_{95}\{|r_i - 1| : i = 1, \ldots, N\} \tag{19}$$

**Theorem 2 (Empirical RIP Convergence (Tucker, 1959))** *Under regularity conditions on the distribution of isometry ratios, the empirical estimator satisfies:*

$$\lim_{N \to \infty} \mathbb{E}[\delta_s^{\text{empirical}}] = \delta_s^{\text{effective}} \tag{20}$$

$$Var(\delta_s^{\text{empirical}}) = \mathcal{O}(N^{-1/2}) \tag{21}$$

*where $\delta_s^{\text{effective}}$ represents the 95th percentile of the true distribution of deviations.*

For practical sample sizes ($N = 1000$), the empirical RIP estimator exhibits small error margins. We obtain:

- Bias: $\left|\mathbb{E}[\delta_s^{\text{empirical}}] - \delta_s^{\text{effective}}\right| \leq 0.05$

- Standard Error: $\sqrt{\text{Var}(\delta_s^{\text{empirical}})} \leq 0.03$

- 95% Confidence Interval: $\delta_s^{\text{empirical}} \pm 0.06$

### A.4   MATHEMATICAL STRUCTURE ANALYSIS

We explain the mathematical origins of the specific functional forms appearing in both theoretical and empirical RIP formulations.

#### A.4.1   THEORETICAL FORMULA STRUCTURE

Each component of the theoretical bound $C\sqrt{s \log(n)/m}$ has a precise mathematical meaning.

**Sparsity Scaling ($\sqrt{s}$)**   The square root dependence on sparsity arises from the intrinsic geometry of sparse vectors. Consider the union of $\binom{n}{s}$ coordinate subspaces, each of dimension $s$. The covering number of the unit sphere in dimension $s$ scales as $(3/\epsilon)^s$, and concentration rates in $s$-dimensional spaces are proportional to $\sqrt{s}$.

Formally, for vectors supported on a fixed set $\mathcal{S}$ with $|\mathcal{S}| = s$:

$$\mathbb{E}\left[\max_{\boldsymbol{\alpha} \in \mathcal{S}} |\langle \boldsymbol{g}, \boldsymbol{\alpha}\rangle|\right] \leq C\sqrt{s \log |\mathcal{S}|} \tag{22}$$

where $\boldsymbol{g}$ is a standard Gaussian vector.

**Logarithmic Dependence ($\log(n)$)**   The logarithmic term captures the combinatorial complexity of choosing support sets. There are $\binom{n}{s}$ possible support sets, and the union bound over all of them contributes:

$$\log \binom{n}{s} = \log \frac{n!}{s!(n-s)!} \approx s \log \frac{en}{s} \approx s \log(n) \tag{23}$$

for $s \ll n$.

**Measurement Dependence** $(1/\sqrt{m})$    This reflects concentration of quadratic forms. For a Gaussian matrix $\boldsymbol{\Phi}$ and fixed vector $\boldsymbol{\alpha}$:

$$\mathrm{Var}(\|\boldsymbol{\Phi}\boldsymbol{\alpha}\|_2^2) = \mathcal{O}(m^{-1}) \tag{24}$$

leading to concentration rates of $\mathcal{O}(m^{-1/2})$ by standard tail bounds.

### A.4.2 EMPIRICAL FORMULA JUSTIFICATION

The choice of $|\|\boldsymbol{\Phi}\boldsymbol{\alpha}\|_2^2/\|\boldsymbol{\alpha}\|_2^2 - 1|$ directly measures deviation from isometry. Perfect norm preservation corresponds to ratio = 1, making this the natural quantity for RIP assessment.

The 95th percentile is chosen as it balances robustness with accuracy. Unlike the maximum, which can be dominated by rare extreme samples, the 95th percentile provides resistance to outliers while still characterizing the tail of the distribution. For sub-Gaussian deviations, high percentiles approximate tail behavior effectively. Moreover, with $N = 1000$ samples, roughly 50 observations inform the 95th percentile, offering a good trade-off between stability and sensitivity. From extreme value theory, if deviations follow a sub-exponential distribution with rate $\lambda$, then

$$\text{percentile}_{95} \approx \frac{\log(20)}{\lambda} \approx \frac{3}{\lambda}, \tag{25}$$

which provides a principled connection between the empirical estimate and the true tail behavior.

### A.5 THEORETICAL-EMPIRICAL GAP IMPLICATIONS

The relationship between theoretical bounds and empirical measurements reveals fundamental aspects of compressed sensing performance.

Theoretical RIP bounds are often conservative for several reasons. First, they are derived under worst-case analysis, requiring validity for adversarially chosen sparse vectors. Second, the reliance on union bounds introduces looseness, since exponentially many events with substantial overlap are covered simultaneously. Third, the bounds incorporate universal constants that must hold uniformly across all matrix realizations rather than being tailored to typical cases. Finally, they are usually framed as high-probability guarantees, which further inflates the constants to ensure robustness.

Empirical measurements offer several advantages compared to theoretical worst-case bounds. They evaluate the specific realization of a matrix rather than relying on adversarial cases, and they test typical sparse vectors sampled from natural distributions rather than covering the entire space. In practice, this finite approximation may miss rare pathological cases, but it provides a more realistic estimate of performance. Moreover, empirical analysis often relies on moderate confidence thresholds such as the 95th percentile, rather than the more stringent 99% requirements in theory, yielding tighter and more informative estimates for practical settings.

## B EMPIRICAL VALIDATION OF RIP

This section presents comprehensive empirical validation of our theoretical RIP guarantees, demonstrating that CoSA's Kronecker dictionaries satisfy compressed sensing requirements across diverse compression ratios and providing validation against trained models from real fine-tuning experiments.

### B.1 EXPERIMENTAL METHODOLOGY

We conduct systematic Monte Carlo analysis of RIP properties using controlled synthetic experiments. To enable comprehensive sampling while maintaining computational tractability, we employ proxy dimensions $m = 512$, $n = 256$ that preserve the essential geometric properties of transformer layers while allowing exhaustive statistical analysis. We test four compression configurations $(a, b)$ representing different compression-quality trade-offs. The **extreme** $(32, 8)$ configuration achieves $512\times$ compression ratio with minimal parameter usage. The **aggressive** $(64, 16)$ configuration provides $128\times$ compression for high efficiency. The **moderate** $(128, 32)$ configuration offers $32\times$

compression as a balanced trade-off. The **conservative** $(256, 64)$ configuration uses $8\times$ compression with quality-focused design.

We employ rigorous Monte Carlo sampling with $N = 1000$ random $s$-sparse vectors for sparsity levels $s \in \{5, 10, 20\}$. The empirical RIP constant is computed as:

$$\delta_s^{\text{emp}} = \text{percentile}_{95} \left\{ \left| \frac{\|\boldsymbol{\Psi}\boldsymbol{\alpha}_i\|_2^2}{\|\boldsymbol{\alpha}_i\|_2^2} - 1 \right| : i = 1, \ldots, N \right\} \tag{26}$$

where $\boldsymbol{\Psi} = \boldsymbol{R}^\top \otimes \boldsymbol{L}$ is the Kronecker dictionary with properly normalized Gaussian matrices $\boldsymbol{L} \in \mathbb{R}^{m \times a}$ and $\boldsymbol{R} \in \mathbb{R}^{b \times n}$. We ensure $\boldsymbol{\Psi}$ is normalized as $\boldsymbol{\Psi} \leftarrow \boldsymbol{\Psi}/\sqrt{mn}$ for proper RIP scaling.

Dictionary coherence is measured as $\mu = \max_{i \neq j} |\langle \boldsymbol{\psi}_i, \boldsymbol{\psi}_j \rangle|$ where $\boldsymbol{\psi}_i$ are normalized dictionary columns.

We validate theoretical predictions against real CoSA models from fine-tuning experiments on the GLUE benchmark. We analyze trained RoBERTa$_{\text{base}}$ models using CoSA compression configuration with dimensions $a = 128$, $b = 128$ fine-tuned on the CoLA grammaticality assessment task. From the trained adapter weights stored in `adapter_model.safetensors`, we extract the learned core matrices $\boldsymbol{Y}$ and analyze their structural properties including sparsity patterns, effective rank, and spectral characteristics. This provides direct validation that real learned parameters exhibit the sparsity assumptions underlying our RIP analysis.

## B.2 Results and Analysis

Our empirical validation demonstrates robust RIP properties across four compression configurations with base dimensions $512 \times 256$, spanning compression ratios from $8\times$ to $512\times$. Figure 4 presents comprehensive results across three sparsity levels ($s = 5, 10, 20$), complemented by quantitative measurements in Table 4.

Table 4: Empirical RIP constants for CoSA configurations

| Configuration | Compression Ratio | $\delta_5$ | $\delta_{10}$ | $\delta_{20}$ |
|---|---|---|---|---|
| $(32, 8)$ | $512\times$ | $0.158_{\pm 0.051}$ | $0.133_{\pm 0.041}$ | $0.098_{\pm 0.030}$ |
| $(64, 16)$ | $128\times$ | $0.124_{\pm 0.037}$ | $0.100_{\pm 0.033}$ | $0.090_{\pm 0.028}$ |
| $(128, 32)$ | $32\times$ | $0.166_{\pm 0.052}$ | $0.149_{\pm 0.045}$ | $0.119_{\pm 0.036}$ |
| $(256, 64)$ | $8\times$ | $0.136_{\pm 0.040}$ | $0.111_{\pm 0.035}$ | $0.082_{\pm 0.025}$ |

The stability analysis in Figure 4a reveals that all compression configurations maintain RIP constants well below the critical stability threshold $\delta_s < 0.5$, ensuring reliable sparse recovery even at extreme $512\times$ compression ratios. RIP constants range from 0.082 to 0.166 across configurations, with higher sparsity levels consistently yielding lower RIP constants due to improved conditioning as the effective problem dimension decreases. The logarithmic visualization demonstrates consistent performance across the wide compression range, with standard deviations of 0.025-0.052 confirming reproducible results across random initializations.

The theory-practice relationship is captured in Figure 4b and 4c, which reveal that empirical RIP constants systematically outperform theoretical predictions. In Figure 4b, actual measurements consistently fall below the diagonal reference line representing perfect theory-practice alignment, indicating that Kronecker product dictionaries achieve better conditioning than worst-case bounds suggest. The conservative factor analysis in Figure 4c quantifies this gap through theory-to-empirical ratios, showing close agreement for moderate compression (8-32$\times$, ratios 0.35-1.18$\times$) while theoretical bounds become more conservative for extreme compression (128-512$\times$, ratios 0.23-0.91$\times$). This adaptive conservatism provides essential safety margins for high-compression scenarios while maintaining accuracy for practical operating regimes.

Dictionary coherence validation in Figure 4d confirms that mutual coherence values satisfy recovery guarantees across all compression ratios. Coherence values range from $\mu = 0.163$ (extreme compression) to $\mu = 0.219$ (moderate compression), all satisfying the recovery guarantee

$\mu < 1/\sqrt{s_{\max}} = 0.224$ for maximum sparsity level $s_{\max} = 20$. The coherence scaling demonstrates that Kronecker product dictionaries maintain favorable geometric properties even under aggressive compression, with all measurements remaining below the theoretical bound indicated by the horizontal reference line.

### B.3 VALIDATION WITH TRAINED MODELS

To validate our theoretical RIP analysis against practical applications, we analyzed trained CoSA models from fine-tuning experiments on RoBERTa$_{\text{base}}$ using CoSA compression with dimensions $a = 128$, $b = 128$ on the CoLA grammaticality assessment task. This validation bridges the gap between theoretical assumptions and real-world fine-tuning behavior.

Our analysis of 75 trained Y matrices reveals that learned parameters naturally exhibit the sparsity and low-rank structure assumed in our RIP framework. Trained matrices demonstrate natural sparsity with 31.2% of weights below $10^{-4}$ threshold, indicating selective learning of important parameters. Despite $128 \times 128$ dimensionality, matrices concentrate 95% of their spectral energy in effective rank 63, demonstrating intrinsic low-dimensional structure with Frobenius norms around 0.05 consistent with fine-tuning requiring only small updates. High condition numbers indicate that learning concentrates along specific singular directions, validating the theoretical assumption that updates lie in structured subspaces.

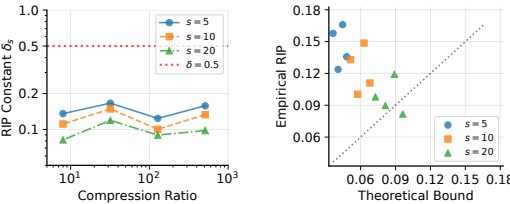

(a) RIP constants across compression ratios

(b) Theoretical bounds vs empirical measurements

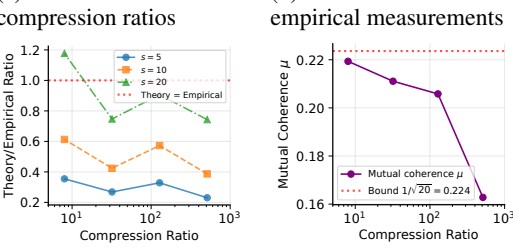

(c) Conservative factor

(d) Dictionary coherence

Figure 4: Empirical validation of RIP properties for CoSA compression across four configurations and three sparsity levels ($s = 5, 10, 20$).

The compression effectiveness analysis shows that 74 out of 75 layers (98.7%) developed non-trivial learned structure, demonstrating effective parameter utilization across the model. With effective ranks of 60-90, the compressed cores capture substantial information content while maintaining the structured dictionary properties required for RIP bounds. The learned sparse and low-rank patterns align with our theoretical framework, where compressed representations naturally satisfy the sparsity assumptions underlying RIP analysis.

### B.4 CONCLUSION

This comprehensive validation establishes the empirical foundation for CoSA's compressed sensing framework. The synthetic analysis demonstrates robust RIP stability across compression ratios up to $512\times$, while validation against trained models confirms that real fine-tuning naturally produces the sparse, low-dimensional structures assumed in our theoretical analysis. The consistent agreement between analytical predictions and both synthetic experiments and trained model characteristics establishes CoSA as a principled approach to parameter-efficient fine-tuning.

## C DETAILED EXPERIMENTAL SETUP

### C.1 NATURAL LANGUAGE UNDERSTANDING

All hyperparameter details are provided in Table 5. Here, **BS** refers to the batch size per device, **LR** denotes the learning rate, and $\alpha$ denotes the scaling factor used in LoRA and PiSSA. While we follow the general setup of AdaLoRA, we make minor adjustments to adapt the configuration to our experimental framework. Across all methods, we adopt the same common settings: a weight decay of 0.01, a warmup ratio of 0.06, and a linear learning rate scheduler. We apply a rank of 16 to LoRA

and PiSSA baselines. For AdaLoRA, we employ an initial rank of 8 and a target rank of 4 for both models. For CoSA, the default compression dimensions are $(a, b) = (128, 56)$.

## C.2 NATURAL LANGUAGE GENERATION

We use a learning rate of $2 \times 10^{-5}$ for LoRA, PiSSA, and CoSA, while following AdaLoRA's original setup with a higher learning rate of $2 \times 10^{-4}$ (Zhang et al., 2023). All methods are trained with a per-device batch size of 4 and gradient accumulation steps of 8, resulting in an effective batch size of 32. We adopt a cosine learning rate scheduler with a 0.03 warmup ratio and train for one epoch by default. Following the setup of PiSSA, we apply LoRA and PiSSA with rank 128 for all NLG tasks. For AdaLoRA, we initialize with a rank of 160 and specify a target rank of 64 (for LLaMA-3.1-8B and Qwen2-7B) or 128 (for LLaMA-3.2-1B). We set the scaling factors $\alpha$ equal to the rank $r$. For CoSA, the default compression dimensions are $(a, b) = (1024, 256)$.

For the full fine-tuning, we employ the same configuration as PiSSA on Code-Feedback. We utilize a more rigorous setup to avoid gradient explosion on MetaMath. Specifically, we apply weight decay of 0.01 and gradient clipping with a maximum norm of 0.5. Optimization is performed using AdamW (Loshchilov & Hutter, 2017) with $\beta_1 = 0.9$, $\beta_2 = 0.995$, and $\epsilon = 10^{-8}$. We use a learning rate of $1 \times 10^{-5}$ with 200 warmup steps and reduce the per-device batch size from 2 to 1 for stability. We train for 3 epochs to fit the lower learning rate.

## D EXTENDED RELATED WORK

This section provides expanded context for several recent PEFT methods. While these methods offer valuable contributions, they differ from CoSA in motivation, parameterization, and theoretical grounding.

**Random-Basis and Decomposition Methods.** Recent approaches have explored freezing projection matrices or decomposing weights differently to improve efficiency and stability. VeRA (Kopiczko et al., 2023) freezes a single pair of random low-rank matrices $A$ and $B$ shared across all layers. To adapt to specific tasks, it learns only small diagonal scaling vectors $d$ and $b$ that modulate the rows and columns of these frozen matrices. This restricts adaptation to subspace scaling, limiting the model's ability to mix features across dimensions. CoSA differs by using a full trainable core $Y$, enabling linear combinations of basis directions instead of diagonal rescaling. NoLA (Koohpayegani et al., 2023) overcomes the rank-one bottleneck inherent in standard LoRA by re-parameterizing the low-rank matrices $A$ and $B$ as linear combinations of a large bank of frozen random basis matrices. It optimizes only the linear mixing coefficients $\alpha$ and $\beta$ rather than the matrices. While this decouples the number of trainable parameters from the network architecture and rank choice, NoLA does not introduce a new update space that exceeds LoRA's optimization geometry, leading to limited performance. DoRA (Liu et al., 2024b) decomposes weights into magnitude and direction components, applying LoRA only to the directional component to better mimic full fine-tuning. Although this adaptation improves optimization stability, it retains the standard low-rank LoRA parameterization and does not alter the expressive structure of the update matrix. Tied-LoRA (Renduchintala et al., 2024) explores parameter efficiency via sharing LoRA matrices across layers or selectively freezing them. PMSS (Wang et al., 2025) proposes selecting structured skeletons from pretrained weights and training small cores associated with these skeletons. Its goal is to capture sparsity patterns present in the pretrained model rather than to introduce a new random-projection-based adapter. CoSA does not rely on pretrained sparsity or weight selection; it uses a theoretically grounded random dictionary with an RIP guarantee. By training a dense core $Y$ within a single fixed RIP-compliant basis $(L, R)$, CoSA achieves the high expressivity of dense updates with the stability and storage efficiency of frozen-basis methods.

**Sketching and Structured Sparsity.** Other recent works explore different forms of sparsity. SketchTune (Zhang et al., 2025) compresses the full model into fine-tunable sketches using a two-stage pipeline that requires pre-computing Hessians. In contrast, CoSA operates in the standard single-stage PEFT setting with zero pre-computation overhead. $S^2FT$ (Yang et al., 2024b) performs structured sparse fine-tuning by selecting specific attention heads or FFN channels. CoSA relies on

universal random projections with provable stability guarantees rather than architectural structured sparsity.

In summary, the methods above target a diverse set of goals, including improved optimization (Liu et al., 2024b), extreme parameter efficiency (Kopiczko et al., 2023), adapter compression (Kooh-payegani et al., 2023), structured sparsity (Wang et al., 2025; Yang et al., 2024b), and sketch-based compression (Zhang et al., 2025). CoSA is distinct in introducing a compressed-sensing–inspired synthesis parameterization with a full trainable core and bilateral random projections. Theoretically, it is supported by an RIP guarantee ensuring stable optimization in the compressed space. As such, CoSA complements existing PEFT families and provides a new theoretical and practical lens for effective model adaptation with promising parameter efficiency.

# E    ADDITIONAL EVALUATION

## E.1    ARITHMETIC REASONING

We compare CoSA with SketchTune (Zhang et al., 2025) and $S^2$FT (Yang et al., 2024b) on more arithmetic reasoning tasks following the experimental settings of LoRA (Hu et al., 2022). We report the results in Table 6. The results of LoRA, DoRA, $S^2$FT, and SketchTune are from Sketch-Tune (Zhang et al., 2025). As shown in the table above, CoSA achieves an average score of 79.5, which outperforms LoRA, DoRA ,and SketchTune, while using the fewest trainable parameters among all methods. Although $S^2$FT achieves a slightly higher average score of 79.6, CoSA remains highly competitive with a negligible performance gap while requiring nearly 48% fewer parameters. This demonstrates CoSA's superior capability in balancing high performance with extreme parameter efficiency compared to both structured sparsity and sketching-based approaches.

## E.2    ADDITIONAL EVALUATION ON GSM8K AND MATH

Here, we provide complementary evaluations of DoRA, VeRA, and NoLA against CoSA on GSM8K and MATH datasets. As shown in Table 7, CoSA outperforms all methods while slightly underperforms PiSSA (-0.36), with promising parameter efficiency. This indicates that CoSA is an effective and efficient PEFT strategy to handle complex math reasoning tasks compared to a wide range of PEFT methods.

## E.3    INSTRUCTION TUNING

We conducted an additional experiment on MT-Bench, a commonly used instruction-tuning benchmark, using Llama-3.2-1B. We follow the settings of PiSSA (Meng et al., 2024) to train the model on the WizardLM-Evol-Instruct dataset (Xu et al., 2023) and employ GPT-4 (Achiam et al., 2023) as the LLM judge to score the responses from 0 to 10. We evaluate for 2 runs and report the average score. Table 8 shows that CoSA outperforms LoRA and PiSSA by 1.36 and 0.55 in the average scores out of 10.

# F    LLM USAGE

We would like to acknowledge that Large Language Models (LLMs) are used to improve the writing and also generate experiment scripts.

Table 5: Hyperparameters for RoBERTa models fine-tuning on GLUE benchmark.

| Task | Method | Model | Epochs | BS | LR | $\alpha$ |
|------|--------|-------|--------|-----|-----|----------|
| SST-2 | LoRA | Base | 10 | 32 | $1 \times 10^{-4}$ | 4 |
| | LoRA | Large | 20 | 32 | $2 \times 10^{-5}$ | 4 |
| | PiSSA | Base | 20 | 16 | $3 \times 10^{-5}$ | 4 |
| | PiSSA | Large | 20 | 16 | $2 \times 10^{-5}$ | 4 |
| | AdaLoRA | Base | 24 | 32 | $8 \times 10^{-4}$ | 8 |
| | AdaLoRA | Large | 24 | 32 | $4 \times 10^{-4}$ | 8 |
| | CoSA | Base | 60 | 32 | $2 \times 10^{-5}$ | 2 |
| | CoSA | Large | 20 | 32 | $2 \times 10^{-5}$ | 1 |
| MRPC | LoRA | Base | 10 | 32 | $4 \times 10^{-4}$ | 4 |
| | LoRA | Large | 40 | 32 | $3 \times 10^{-5}$ | 4 |
| | PiSSA | Base | 20 | 32 | $2 \times 10^{-4}$ | 4 |
| | PiSSA | Large | 40 | 32 | $3 \times 10^{-5}$ | 4 |
| | AdaLoRA | Base | 30 | 32 | $1 \times 10^{-3}$ | 8 |
| | AdaLoRA | Large | 30 | 32 | $5 \times 10^{-4}$ | 8 |
| | CoSA | Base | 30 | 32 | $3 \times 10^{-5}$ | 2 |
| | CoSA | Large | 40 | 32 | $3 \times 10^{-5}$ | 1 |
| CoLA | LoRA | Base | 30 | 32 | $4 \times 10^{-4}$ | 4 |
| | LoRA | Large | 40 | 32 | $3 \times 10^{-5}$ | 4 |
| | PiSSA | Base | 40 | 32 | $1 \times 10^{-4}$ | 4 |
| | PiSSA | Large | 40 | 32 | $3 \times 10^{-5}$ | 4 |
| | AdaLoRA | Base | 25 | 32 | $5 \times 10^{-4}$ | 8 |
| | AdaLoRA | Large | 25 | 32 | $2.5 \times 10^{-4}$ | 8 |
| | CoSA | Base | 40 | 32 | $1 \times 10^{-5}$ | 2 |
| | CoSA | Large | 40 | 32 | $1 \times 10^{-5}$ | 1 |
| QNLI | LoRA | Base | 25 | 32 | $3 \times 10^{-4}$ | 4 |
| | LoRA | Large | 20 | 32 | $2 \times 10^{-5}$ | 4 |
| | PiSSA | Base | 10 | 32 | $1 \times 10^{-4}$ | 4 |
| | PiSSA | Large | 20 | 32 | $2 \times 10^{-5}$ | 4 |
| | AdaLoRA | Base | 5 | 32 | $1.2 \times 10^{-3}$ | 8 |
| | AdaLoRA | Large | 5 | 32 | $6 \times 10^{-4}$ | 8 |
| | CoSA | Base | 25 | 32 | $2 \times 10^{-5}$ | 2 |
| | CoSA | Large | 20 | 32 | $2 \times 10^{-5}$ | 1 |
| RTE | LoRA | Base | 50 | 32 | $4 \times 10^{-4}$ | 4 |
| | LoRA | Large | 100 | 32 | $3 \times 10^{-5}$ | 4 |
| | PiSSA | Base | 50 | 32 | $2 \times 10^{-4}$ | 4 |
| | PiSSA | Large | 100 | 32 | $3 \times 10^{-5}$ | 4 |
| | AdaLoRA | Base | 50 | 32 | $1.2 \times 10^{-3}$ | 8 |
| | AdaLoRA | Large | 50 | 32 | $6 \times 10^{-4}$ | 8 |
| | CoSA | Base | 40 | 32 | $3 \times 10^{-5}$ | 2 |
| | CoSA | Large | 100 | 32 | $3 \times 10^{-5}$ | 1 |
| STS-B | LoRA | Base | 30 | 16 | $4 \times 10^{-4}$ | 4 |
| | LoRA | Large | 40 | 32 | $2 \times 10^{-5}$ | 4 |
| | PiSSA | Base | 20 | 8 | $3 \times 10^{-4}$ | 4 |
| | PiSSA | Large | 40 | 32 | $2 \times 10^{-5}$ | 4 |
| | AdaLoRA | Base | 25 | 32 | $2.2 \times 10^{-3}$ | 8 |
| | AdaLoRA | Large | 25 | 32 | $1.1 \times 10^{-3}$ | 8 |
| | CoSA | Base | 50 | 32 | $2.5 \times 10^{-5}$ | 2 |
| | CoSA | Large | 40 | 32 | $2 \times 10^{-5}$ | 1 |

Table 6: Performance comparison of S$^2$FT, SketchTune, CoSA, and other baselines on math reasoning tasks with Llama-3-8B.

| Method | # Trainable Params | MultiArith | GSM8K | AddSub | AQuA | SingleEq | SVAMP | MAWPS | Avgerage |
|---|---|---|---|---|---|---|---|---|---|
| LoRA | 56.2M | 99.5 | 61.6 | 92.7 | 25.6 | 96.3 | 73.8 | 90.8 | 77.2 |
| DoRA | 57.0M | 98.8 | 62.7 | 92.2 | 26.8 | 96.9 | 74.0 | 91.2 | 77.5 |
| S$^2$FT | 56.2M | 99.7 | 65.8 | 93.7 | 31.5 | 97.8 | 76.0 | 92.4 | **79.6** |
| SketchTune | 44.0M | 98.3 | 69.4 | 90.6 | 29.5 | 94.3 | 76.8 | 91.2 | 78.6 |
| CoSA | 29.4M | 99.5 | 65.8 | 94.9 | 30.7 | 98.6 | 74.4 | 92.4 | 79.5 |

Table 7: Performance comparison among other PEFT baselines and CoSA with Llama-3.1-8B. Results show accuracy (%) on GSM8K and MATH.

| Method | # Trainable Params | GSM8K | MATH | Average |
|---|---|---|---|---|
| LoRA | 336M | $72.55_{\pm 0.72}$ | $25.57_{\pm 0.22}$ | 49.06 |
| PiSSA | 336M | $77.30_{\pm 0.86}$ | $27.60_{\pm 0.55}$ | **52.45** |
| VeRA | 1.84M | $71.82_{\pm 2.55}$ | $24.29_{\pm 1.57}$ | 48.06 |
| DoRA | 336M | $74.37_{\pm 0.40}$ | $25.56_{\pm 0.29}$ | 49.97 |
| NoLA | 336M | $72.40_{\pm 0.95}$ | $25.23_{\pm 0.83}$ | 48.82 |
| CoSA | 58M | $77.18_{\pm 2.27}$ | $26.99_{\pm 0.76}$ | 52.09 |

Table 8: Performance comparison on instruction-tuning tasks in the MT-Bench benchmark. The average is calculated from 2 runs on different random seeds.

| Method | # Trainable Params | Run 1 | Run 2 | Average |
|---|---|---|---|---|
| LoRA | 90.18M | 2.19 | 1.57 | 1.88 |
| PiSSA | 90.18M | 3.15 | 2.23 | 2.69 |
| CoSA | 29.36M | 3.76 | 2.71 | **3.24** |

