# OpenReview forum: "CoSA: Compressed Sensing-Based Adaptation of Large Language Models"
_ICLR.cc/2026/Conference — ICLR 2026 Conference Withdrawn Submission_

### Official Review · Reviewer_jcn4 · 2025-10-27

**Soundness:** 2
**Presentation:** 2
**Contribution:** 1
**Rating:** 2
**Confidence:** 4

**Summary:**

This paper proposes CoSA (Compressed Sensing Adapter), a compressed sensing-based adapter architecture for parameter-efficient fine-tuning. Unlike low-rank approaches, CoSA expresses weight updates in a compressed form using a learnable core matrix \( Y \) and fixed random projection matrices \( L \) and \( R \). The method is theoretically motivated by the Restricted Isometry Property (RIP), which is used to justify both training stability and the preservation of expressivity.

**Strengths:**

- The idea of applying compressed sensing to parameter-efficient fine-tuning is interesting.
- The paper is clearly written and easy to follow.
- There is an effort to ground the method in compressed sensing theory, including the use of the Restricted Isometry Property (RIP).

**Weaknesses:**

- The proposed method lacks novelty. Several works have already explored tri-matrix adapter structures. In particular, TLoRA [1] (in arxiv) presents a structurally identical approach, using frozen random matrices $A$ and $C$, and a learnable small matrix $B$.  Additionally, PMSS [2], which trains frozen A, B, and learnable cores in the same way, but with different initialization methods, was proposed in COLING'25. The authors did not provide a sufficient comparison of these tri-matrix adapters.

- Although the paper claims that RIP leads to stable training, it is unclear whether such constraints are always beneficial. Since RIP restricts the amount of change after projection, it could potentially limit the expressivity of the model during fine-tuning, especially in low-data or few-shot scenarios where greater flexibility might be required. For instance, LoRA-GA [2] aims to better approximate full fine-tuning by mimicking full gradients, while CoSA instead constrains the update space, which may actually hinder learning. While the paper emphasizes the benefits of RIP, it lacks concrete theoretical, empirical, or quantitative evidence to support this claim.

- The set of baseline comparisons is too narrow. The paper does not compare CoSA against recently proposed methods such as NoLA [3] and VeRA [4], which also use frozen/random bases or $AB$-structured adapters. In addition, there is no experimental comparison with TLoRA, which appears to be the most closely related work. Such comparisons are essential to fairly assess CoSA’s effectiveness.

- Unlike standard LoRA, CoSA introduces \( ab \) parameters. Therefore, to ensure fair comparison with LoRA, the number of trainable parameters should be explicitly reported in each experiment. For example, in the NLG task with LLaMA, the paper mentions using (a, b) = (1024, 256), while the LoRA rank is set to 128. Since LLaMA-3B has a hidden dimension of 2048, this results in:
    - LoRA: 2048 × 128 × 2 parameters
    - CoSA: 1024 × 256 parameters

Therefore, CoSA uses about half the parameters compared to LoRA. However, other methods such as VeRA [4] or Vb-LoRA [5] introduce even fewer parameters, which makes CoSA to be less contributed. Therefore, the paper should clearly report the exact parameter counts and include comparisons against a wider range of parameter-efficient baselines.



---

>[1]Islam, Tanvir. "TLoRA: Tri-Matrix Low-Rank Adaptation of Large Language Models." arXiv preprint arXiv:2504.18735 (2025).
>
>[2] PMSS: Pretrained Matrices Skeleton Selection for LLM Fine-tuning, COLING, 2025
>
>[3]Wang, Shaowen, Linxi Yu, and Jian Li. "Lora-ga: Low-rank adaptation with gradient approximation." Advances in Neural Information Processing Systems 37 (2024): 54905-54931.
>
>[4] Koohpayegani, Soroush Abbasi, et al. "NOLA: Compressing LoRA using Linear Combination of Random Basis." The Twelfth International Conference on Learning Representations. 2024
>
>[5] Kopiczko, Dawid Jan, Tijmen Blankevoort, and Yuki M. Asano. "VeRA: Vector-based Random Matrix Adaptation." The Twelfth International Conference on Learning Representations. 2024
>
>[6] Li, Yang, Shaobo Han, and Shihao Ji. "Vb-lora: Extreme parameter efficient fine-tuning with vector banks." Advances in Neural Information Processing Systems 37 (2024): 16724-16751.

**Questions:**

- In Table 1, the paper claims that the storage requirement is $\mathcal{O}(1)$, but the learned core matrix $Y \in R^{a \times b}$ still needs to be stored. To me, this suggests that the storage complexity should be $\mathcal{O}(ab)$, not $\mathcal{O}(1)$. If I’m misunderstanding something, clarification would be appreciated. Moreover, in the NLG task, it seems that $a, b$ can be as large as 1024, which is not negligible in practice.

---

> ### Author Response · Authors · 2025-11-24
> **Author Rebuttal: Part 1**
>
> **Weakness 1: Lack of novelty. The authors did not provide a sufficient comparison of these tri-matrix adapters.**
>
> We appreciate the reviewer for pointing out these related works. Although our method shares a high-level tri-matrix topology with them, we would like to clarify that this structural similarity is superficial. CoSA is fundamentally different, and our novelty lies in both its theoretical grounding and practical flexibility.
> * TLoRA and PMSS operate strictly within the low-rank LoRA family. In TLoRA, the matrices $A \in \mathbb{R}^{k \times r}$ and $A \in \mathbb{R}^{r \times d}$ impose a hard rank-r bottleneck. Its rank is $\text{rank}(ABC) \le r$ regardless of how $B$ is initialized or trained. This matches LoRA’s rank-r update formulation and aligns with TLoRA's inferior performance against standard LoRA (shown in Table 3 of the TLoRA paper). PMSS likewise focuses on selecting sparse *skeletons* from pretrained weights and does not employ bilateral random projections, which may also be constrained by the prior knowledge if the base model is not strong enough. These approaches do not offer a compressed-sensing interpretation, nor do they provide guarantees on the geometry of the update space. In contrast, CoSA is built around a compressed-sensing synthesis model. We prove that the induced Kronecker dictionary $R^\top \otimes L$ satisfies a Restricted Isometry Property (RIP) with high probability. This yields a theoretically justified embedding that preserves geometry in the reduced parameter space, which is absent in TLoRA and PMSS.
> * CoSA offers greater practical flexibility. The expressive capacity is controlled by freely choosing $a$ and $b$. We empirically demonstrate that asymmetrically choosing $(a,b)$ usually enhances performance over $(r,r)$ in Section 5.3.1. Moreover, CoSA requires storing only the small core $Y$ and a random seed to regenerate $L$ and $R$, making it lightweight in memory and easy to deploy. This flexibility and efficiency are not available in TLoRA or PMSS.
>
> **Weakness 2: It is unclear whether such constraints are always beneficial. Since RIP restricts the amount of change after projection, it could potentially limit the expressivity of the model during fine-tuning, especially in low-data or few-shot scenarios where greater flexibility might be required. LoRA-GA [2] aims to better approximate full fine-tuning by mimicing full gradients, while CoSA instead constrains the update space, which may actually hinder learning. The paper lacks concrete theoretical, empirical, or quantitative evidence to support this claim.**
>
> * Theoretical evidence:
>     * We would like to clarify that our use of RIP does not impose a constraint on the expressivity of the update. Instead, it guarantees stability of optimization in the reduced parameter space.
>     * From Theorem 1 and Equation 8 in the CoSA paper, RIP ensures the mapping from $Y$ to $\Delta W = LYR$ is an isometry and stable. For example, the gradient directions in the compressed space do not collapse or explode while being mapped to the high-dimensional space.
>     * In CoSA, expressivity is controlled solely by the number of trainable parameters $ab$ instead of RIP. The Kronecker product of $L$ and $R$ serves as the projection dictionary, and the dimension of $Y$ determines the probability of satisfaction of RIP. Our experiments show that CoSA improves accuracy across various datasets relative to LoRA and PiSSA at matched parameter budgets, indicating that RIP-based conditioning does not hinder learning. Instead, it helps CoSA train reliably even under substantial compression. This is also empirically justified in the ablation study of $a$ and $b$.
>     * While LoRA-GA aims to improve convergence by mimicking full fine-tuning gradients via initialization, the robustness of small LLMs with complex reasoning tasks is not investigated or reported. The full fine-tuning of small models can potentially be unstable and prone to overfitting. In CoSA, by compressing the optimization to a stable, isometric subspace, we ensure that the learned updates generalize well independently of the quality of pre-training.
> * Empirical evidence:
>     * We direct attention to our GLUE Benchmark results (Table 2 in the CoSA paper), specifically on the smallest and most sensitive datasets. For example, on the RTE task with only 2.5k training data, CoSA achieves 74.85% accuracy with RoBERTA-Base, outperforming LoRA (72.8%), other baselines, and even full fine-tuning. This improvement is also mainly represented in the RoBERTA-Large results. On another low-data task, MRPC with only 3.7k training samples, CoSA outperforms the baselines, showing its superior adaptation capability on low-resource tasks.
>     * Moreover, in the $(a,b)$ ablation study, we demonstrate that the dimension of $Y$ scales the performance in a reasonable range. This reflects that the reconstruction quality also scales with the probability of satisfaction of RIP.

---

> ### Author Response · Authors · 2025-11-24
> **Author Rebuttal: Part 2**
>
> **Weakness 3: The set of baseline comparisons is too narrow. The paper does not compare CoSA against recently proposed methods such as NoLA [3] and VeRA [4], which also use frozen/random bases or $AB$-structured adapters. In addition, there is no experimental comparison with TLoRA, which appears to be the most closely related work. Such comparisons are essential to fairly assess CoSA’s effectiveness.**
>
> We conduct a new experiment on VeRA and NoLA on the math reasoning benchmark. VeRA is additionally evaluated on the GLUE benchmark for more comparison insights. Moreover, we would like to clarify that TLoRA is not directly comparable to CoSA in practice, while we certainly discuss all three methods in the related-work section.
> * While TLoRA is closely related to CoSA, it is not open-sourced. And the paper does not provide sufficient implementation details (e.g., initialization, sharing scheme, hyperparameters, or training protocol) to reproduce results reliably. Therefore, fairly reproducing it as a baseline is not feasible. The paper also only provide experimental results on a subset of GLUE benchmark showing limited performance, which is not reliable for reference.
> * VeRA parameterizes the update as $\Delta W = \Lambda_d B \Lambda_b A$, where $A$ and $B$ are frozen random matrices and only the diagonal vectors $d$ and $b$ are learned. This effectively restricts the method to a diagonal core, meaning it cannot mix input and output directions. Each input feature from $B$ can only scale a corresponding output direction in $A$.
> CoSA instead uses $\Delta W = L Y R$ with a full trainable matrix $Y$, allowing arbitrary linear combinations of the random basis vectors and far richer transformations. Moreover, VeRA provides no theoretical justification for the conditioning or stability of this diagonal-scaling parameterization, whereas CoSA is grounded in compressed-sensing theory. Our Kronecker dictionary satisfies RIP (Theorem 1), giving a near-isometry guarantee that ensures a well-conditioned optimization landscape.
> * NoLA does not propose a new update space. Instead, it compresses an existing LoRA module by expressing LoRA’s low-rank factors as mixtures of a small set of random basis vectors. As described in the NoLA paper, the adapter remains mathematically equivalent to a LoRA update, and the learned parameters only select or reweight preexisting random bases. Thus, NoLA is a parameter-compression and storage-reduction technique rather than a new parameterization of weight updates. CoSA, in contrast, introduces a different compressed-sensing-motivated synthesis form with a trainable dense core and bilateral random projections. Therefore, NoLA and CoSA address different goals and are not directly competing for the same metric (efficiency vs. performance).
>
> Table 1 in the universal author comment above shows that CoSA consistently outperforms VeRA and NoLA in math reasoning tasks, with an intermediate number of trainable parameters among the three methods. Table 2 shows that CoSA matches or surpasses VeRA with slightly more trainable parameters. These observations indicate that CoSA is a powerful PEFT method and can achieve non-extreme but promising parameter efficiency.
>
> **Weakness 4: Unlike standard LoRA, CoSA introduces (ab) parameters. Therefore, to ensure fair comparison with LoRA, the number of trainable parameters should be explicitly reported in each experiment. For example, in the NLG task with LLaMA, the paper mentions using (a, b) = (1024, 256), while the LoRA rank is set to 128. Since LLaMA-3B has a hidden dimension of 2048, this results in: LoRA: 2048 × 128 × 2 parameters, CoSA: 1024 × 256 parameters**
>
> Formally, CoSA introduces $a \times b$ trainable parameters. We report the number of trainable parameters along with the performance metrics accordingly in the universal tables (Table 1, Table 2, and Table 3). CoSA mostly outperforms other baselines in Vperformance and costs relatively fewer trainable parameters. However, as VeRA and Vb-LoRA can further reduce the number of trainable parameters, their performance is usually limited by the less expressive power. Therefore, we respectfully argue that our goal is to enhance the performance rather than achieve extreme parameter efficiency.

---

> ### Author Response · Authors · 2025-11-24
> **Author Rebuttal: Part 3**
>
> **Question 1: In Table 1, the paper claims that the storage requirement is $\mathcal{O}(1)$, but the learned core matrix $Y \in R^{a \times b}$ still needs to be stored. To me, this suggests that the storage complexity should be $\mathcal{O}(ab)$, not $\mathcal{O}(1)$. If I’m misunderstanding something, clarification would be appreciated. Moreover, in the NLG task, it seems that $a, b$ can be as large as 1024, which is not negligible in practice.**
>
> * We thank the reviewer for pointing out the mispresentation in Table 1 in the original paper. For the storage of $L$ and $R$ matrices, $O(1)$ space is needed because we only need a random seed to generate the projection matrices on-the-fly during inference. For the storage of the trainable core $Y$, we need $O(ab)$ space based on the dimension of $Y$. Overall, we formally need $O(ab)$ to store the whole adapter module. We have corrected Table 1 in the paper accordingly.
> * It is correct that one dimension of $Y$ can be as large as 1024 in the NLG task. However, with $(a,b) = (1024,256)$, CoSA is still able to achieve 58M trainable parameters for *LLama-3.1-8B*, which is less than LoRA (rank=128) with 335M trainable parameters. Under this comparison condition, CoSA outperforms LoRA as shown in Table 3 in the paper.
> * In terms of the concern that the larger $(a,b)$ may not be negligible on NLG tasks, we report the training time in the table below. All training is conducted on the same hardware/software environment as stated in the paper. The table shows that CoSA trains faster than VeRA, AdaLoRA, and DoRA, while slightly slower than PiSSA/LoRA because their $BA$ formulation does not introduce any overhead. All methods remain in the same range of inference latency, around 500 seconds, for all GSM8K and MATH testing samples. Although the training time and latency may be slightly affected by many real-world factors such as machine overhead, we hope the results can provide an intuitive and clear insight that the computational overhead of CoSA is not a concerning issue in practice.
>
> **Table.** Training time (781 total steps) and inference latency of different PEFT methods on MetaMathQA dataset. The number is the average of three runs.
> | Method  | Training Time (minutes) | Inference Latency (seconds)|
> |---------|---------------------|---------------------|
> | LoRA    | 54.2                  |     502              |
> | AdaLoRA | 65.5                  |     502              |
> | PiSSA   | 54.0                  |     505             |
> | NoLA    | 54.2                  |     500             |
> | DoRA    | 68.1                  |     509              |
> | VeRA    | 63.8                  |     517             |
> | CoSA    | 58.4                  |     501

---

### Official Review · Reviewer_bos4 · 2025-10-29

**Soundness:** 2
**Presentation:** 2
**Contribution:** 2
**Rating:** 4
**Confidence:** 5

**Summary:**

The paper proposes Compressed Sensing–based Adaptation (CoSA), a PEFT method for LLMs.

Inspired by compressed sensing thoery, the general idea of this work is to parameterize every weight update as are fixed random projection matrices and only the compact core is trained.

CoSA is compared against LoRA, AdaLoRA, and PiSSA on GLUE and on math/code generation, on top of Llama-3.2-1B, Llama-3.1-8B, Qwen2-7B, showing competitive or better accuracy with substantially fewer parameters.

**Strengths:**

+ This paper is overall well-written and clearly-presented, making the readers easy to follow.

+ The proposed method shows a clear parameter and memory benefits over LoRA, AdaLoRA, PiSSA.

+ The ablation study is extensive.

**Weaknesses:**

- The technique soundness is open to doubt, at least in its current form. For example, the framing is not tied to an actual sparsity prior or to constraints. Besides, there is no theory level proof to justify the stability guarantees.

- The core idea to fix random $L$, $R$ and learn a compact core is not sufficiently distinguished from VeRA and/or other related random-projection PEFT methods, making the contribution to the community difficult to justify.

- This paper does not provide a theory-level proof on the emperical risk bound of either sparisty or the regularized training.

- The compared state-of-the-art PEFT methods are significantly missing. Some more recent and much stronger PEFT methods are mssing for comparison, for example:

[1] DoRA: Weight-Decomposed Low-Rank Adaptation. ICML 2024.

[2] VeRA: Vector-based Random Matrix Adaptation. ICLR 2024.

[3] Foura: Fourier low-rank adaptation. NeurIPS 2024.

[4] SSH: Sparse Spectrum Adaptation via Discrete Hartley Transformation. NAACL 2024.

- In the $(a, b)$ ablation, is the rank $r$ rigorously matched? Please clarify.

- If comparing with these more recent PEFT methods, the performance of the proposed method is rather limited and even inferior.

- The experimental validation, to be honest, is rather limted. The authors only validate on two benchmarks, where GLUE is already out-of-date. It should be benchmarked on more recent yet more challenging instruction-tuning or multi-task mixture benchmarks, like [1-4] do.

- Still regarding the performance, this paper lacks a convincing discussion on the performance variance caused by the random seeds.

- The training time and latency is not either reported or compared.

- Some typos and presentation issues still remain.

**Questions:**

Please refer to the weakness section, and address them point-by-point.

---

> ### Author Response · Authors · 2025-11-24
> **Author Rebuttal: Part 1**
>
> **Weakness 1: The technique soundness is open to doubt, at least in its current form. For example, the framing is not tied to an actual sparsity prior or to constraints. Besides, there is no theory level proof to justify the stability guarantees.**
> We thank the reviewer for this comment, which highlights the need to clarify our specific application of Compressed Sensing (CS) theory. Those concerns can be addressed by the proofs and statements in Section 4.1 and Appendix A.1. Here are the clarifications and explanations:
> * Sparsity: Sparsity can be strictly defined by a matrix with many zero entries or loosely interpreted as a matrix with many small but non-zero numbers. However, in more general cases in CS, signals may not be perfectly sparse but "compressible," meaning they can be well approximated by a small number of non-zero coefficients (Miles et al. 2016). Indeed, CoSA is not tied to literal elementwise sparsity constraints (most entries are zeros). Instead, CoSA follows the synthesis formulation of CS in which sparsity refers to low intrinsic dimensionality in an overcomplete random dictionary. Therefore, it is not necessary to apply sparsity prior or constraints such as $L_1$ penalty to the update matrix $Y$. But it will not hurt the accuracy of the reconstruction of the update matrix. We follow the RIP condition for reconstruction accuracy.
> * Stability guarantees: We respectfully point out that the proof is introduced in Section 4.1. Theorem 1 proves that the Kronecker product dictionary used by CoSA satisfies RIP with high probability. As long as the Restricted Isometry Property (RIP) is satisfied, according to Equation 8 ($||\Psi(\alpha_1 - \alpha_2)||_2^2 \approx ||\alpha_1 - \alpha_2||_2^2$), the reconstruction accuracy and stability are guaranteed. Therefore, the satisfaction of RIP ensures the optimization stability. Furthermore, the stability is reflected in the experimental results that CoSA outperforms baselines with fewer trainable parameters.
>
> [1] Lopes, Miles E. "Unknown sparsity in compressed sensing: Denoising and inference." IEEE Transactions on Information Theory 62.9 (2016): 5145-5166.
>
> **Weakness 2: The core idea to fix random $L$, $R$ and learn a compact core is not sufficiently distinguished from VeRA and/or other related random-projection PEFT methods, making the contribution to the community difficult to justify.**
> Although both CoSA and VeRA use frozen random matrices, CoSA introduces a fundamentally different architectural mechanism. Whereas VeRA performs subspace scaling through two learned vectors, CoSA performs subspace mixing through a compact learned core matrix $𝑌$. This change is not cosmetic: it alters how the model leverages the fixed random projections, enabling richer transformations and addressing expressivity limitations inherent to purely vector-based adaptations. In short, the crucial distinction is how each method exploits the frozen random subspace, and CoSA’s matrix-based mixing provides capabilities that VeRA’s vector-based scaling cannot replicate:
>
> * VeRA ($\Delta W = \Lambda_d B \Lambda_b A$ ) freezes matrices $A$ and $B$ and learns vectors $d$ and $b$ to significanly reduce the number of trainable parameters. Mathematically, this is equivalent to learning a diagonal core. It assumes that the useful entries in the random input projection $B$ align perfectly one-to-one with the necessary output directions $A$. However, it cannot mix feature $i$ from the input with feature $j$ from the output (when $i \neq j$), intrinsically limiting the expressiveness.
> * CoSA ($\Delta W = L Y R$) freezes $L$ and $R$ but learns a non-diagonal matrix $Y$. This allows the model to learn linear combinations of the random basis vectors. The model can take the $i$-th feature from the random input projection $R$ and map it to any combination of directions in the output projection $L$.
>
> Moreover, VeRA lacks a theoretical justification for the geometry of the optimization landscape of updating diagonal scaling vectors and projecting them with random matrices. It does not provide guarantees regarding the stability of training the vectors via gradient descent. In contrast, CoSA is grounded in the synthesis model of Compressed Sensing (CS) theory. By framing the weight update as a signal constructed from a dictionary, we leverage Theorem 1 to show that our specific Kronecker-product basis satisfies the Restricted Isometry Property (RIP). This guarantees that the mapping is a near-isometry (Equation 8), ensuring that the optimization landscape is well-conditioned and that gradients are stable.

---

> ### Author Response · Authors · 2025-11-24
> **Author Rebuttal: Part 2**
>
> **Weakness 3: This paper does not provide a theory-level proof on the emperical risk bound of either sparisty or the regularized training.**
>
> We agree that developing empirical-risk bounds for large-scale LLM fine-tuning would be valuable. However, establishing such bounds remains an open challenge in the PEFT literature. To the best of our knowledge, no existing PEFT method (e.g., LoRA, AdaLoRA, PiSSA, VeRA, DoRA, S²FT) provides excess-risk guarantees for downstream tasks. Prior work instead focuses on heuristic, empirical, or structural motivations.
>
> Our goal in this paper is to provide theory at the level that is currently tractable and meaningful: specifically, guarantees based on the RIP condition that ensure stable optimization in the compressed parameter space (in Theorem 1, Equation 8, and Appendix A.1). As we discussed in the reply on Weakness 1, we do not need explicit regularized training such as $L_1$ penalty to force the literal sparsity of the update matrix. Instead, the satisfaction of RIP provides the geometric guarantee that optimization in the compressed parameter space is stable for any signals. This level of analysis is consistent with the theoretical scope typical for PEFT methods, and we believe it appropriately supports the soundness of our approach.
>
> **Weakness 4: The compared state-of-the-art PEFT methods are significantly missing. Some more recent and much stronger PEFT methods are missing for comparison.**
>
> To make the comparisons with state-of-the-art (SOTA) methods, we demonstrate the comparison results with two SOTA schemes, DoRA and VeRA, in Table 1 in the universal author rebuttal comment. Based on the results, CoSA can outperform these SOTA methods in performance with promising parameter efficiency. We also provide a conceptual discussion on these methods in a universal rebuttal comment. Accordingly, we have added discussions about these methods in the extended related work section in Appendix D.
>
> **Weakness 5: In the (a,b) ablation, is the rank rigorously matched?**
>
> * In the $(a,b)$ ablation, rank $r$ is not rigorously matched as we do not enforce a low-rank constraint as LoRA does. While LoRA’s rank is explicitly bounded by $r=\text{min}(m,n)$, the update matrix $\Delta W = LYR$ in CoSA typically has high effective rank even when $(a,b)$ are small (because $L$ and $R$ are full-rank random matrices). Therefore, we do not match the rank because it is not clearly defined as a hyperparameter, and it does not directly reflect the expressiveness from CoSA's perspective.
> * Instead of using rank, CoSA represents the reconstruction accuracy of $\Delta W$ based on the probability of satisfaction of RIP following Equation 4. The probability is determined by the dimension of $Y$ (a,b). Therefore, the goal of this ablation study is to analyze CoSA's performance scalability while tuning (a,b), rather than comparing with LoRA with a similar trainable parameter budget. In the main experimental results (Table 1 and Table 2), we match (reduce) the number of trainable parameters as a standard practice in PEFT work. Please kindly refer to the table in the universal author comment.
>
> **Weakness 6: If comparing with these more recent PEFT methods, the performance of the proposed method is rather limited and even inferior.**
>
> We thank the reviewer for raising this concern. After adding new results under matched experimental settings, we find that CoSA consistently outperforms both VeRA and DoRA by at least 2.17 points on average on math-reasoning tasks. On NLU tasks, CoSA achieves performance comparable to VeRA on the GLUE benchmark, with only slightly more trainable parameters. These comparisons were conducted by applying all methods under the unified setting (i.e., updating all linear layers) while preserving each method’s original hyperparameters to ensure fairness.
>
> To clarify why our results differ from those reported in the VeRA paper: the two works adopt incompatible experimental setups, making direct cross-paper comparison misleading. VeRA fine-tunes only the Query and Key layers and leaves the classification heads fully trainable, whereas CoSA (following PiSSA’s protocol) updates all linear layers across both NLG and NLU tasks. Additionally, VeRA’s reported trainable-parameter counts exclude classification-head parameters, further inflating the apparent efficiency gap. Under a unified setting, however, CoSA performs competitively or better across benchmarks.

---

> ### Author Response · Authors · 2025-11-24
> **Author Rebuttal: Part 3**
>
> **Weakness 7:The experimental validation, to be honest, is rather limted. The authors only validate on two benchmarks, where GLUE is already out-of-date. It should be benchmarked on more recent yet more challenging instruction-tuning or multi-task mixture benchmarks**
>
> We appreciate the reviewer’s suggestion to include more recent and challenging benchmarks. We would like to clarify that our use of GLUE remains aligned with current practice in the PEFT literature. Recent works such as VeRA, FouRA, SSH, and PiSSA continue to adopt GLUE as the standard NLU benchmark. Thus, while GLUE is a longstanding benchmark, it is still widely used for evaluating the generalization ability of PEFT methods.
>
> In addition, although grouped together in Table 1 of our paper, our evaluation already spans diverse and challenging task types: (HumanEval, MBPP) for code generation, and (GSM8K, MATH) for mathematical reasoning. These benchmark families target distinct capabilities and—especially the coding and math reasoning tasks—represent some of the most difficult domains for small models.
>
> Moreover, we conducted an additional experiment on MT-Bench, a commonly used instruction-tuning benchmark, using Llama-3.2-1B. We report the results in Table 4 in the universal author rebuttal comment. Under this setting, CoSA outperforms LoRA and PiSSA by 1.36 and 0.55 points, respectively (scores out of 10). This provides further evidence that CoSA maintains strong performance on more complex, instruction-following tasks beyond those originally included.
>
> **Weakness 8: Still regarding the performance, this paper lacks a convincing discussion on the performance variance caused by the random seeds.**
>
> We acknowledge the importance of evaluating performance stability. To address this, we have reported the mean and standard deviation across more random seeds ($N=5$). As shown in the table, while there is inherent variance, especially for complex reasoning tasks, CoSA matches the performance of PiSSA while leveraging fewer trainable parameters. PiSSA is more robust to initialization noise because of its SVD-based initialization strategy that leverages the prior knowledge of the base model. In addition, CoSA still outperforms PiSSA on the GLUE benchmark with low variance. As we state in the limitations of CoSA, extending the evaluation to more varied benchmarks remains important future work. We will continue conducting more comprehensive experiments to revise this paper.
>
> **Table.** PiSSA vs. CoSA on math reasoning tasks across 5 runs with Llama-3.1-8B.
> | Method | # Trainable Params | GSM8K | MATH |
> |--------|---------|-------|------|
> | PiSSA  |  336M        |77.18±0.33      |  27.59±0.34    |
> | CoSA   |  58M        |77.18±1.63      |  27.08±0.55    |

---

> ### Author Response · Authors · 2025-11-24
> **Author Rebuttal: Part 4**
>
> **Weakness 9: The training time and latency is not either reported or compared.**
>
> * Latency:
>     * During inference, we merge the adapter weights into the base model. Our method generates the low-rank matrices $L$ and $R$ on the fly using stored random seeds. While this introduces a theoretical overhead, it is computationally negligible with highly optimized random number generation on modern GPUs. On the other hand, CoSA trades a mathematically negligible amount of compute for a massive reduction in storage and memory.
>     * CoSA also allows for the adapter weights to be merged into the base model parameters prior to saving and deployment. Thus, inference latency is identical to the base model and other similar baselines such as LoRA, PiSSA, and so on.
>     * As a result, we report inference latencies in the following table. From the table, all methods remain in the same range of inference latency, around 500 seconds, for all GSM8K and MATH testing samples. Thus, CoSA does not introduce additional inference latency compared to other PEFT methods.
>
> * Training time:
>     * We have now added the training times for LoRA, AdaLoRA, PiSSA, DoRA, VeRA, and CoSA in the table below, using the same experimental setup reported in the paper. The results show that CoSA trains faster than VeRA, AdaLoRA, and DoRA, and is only slightly slower than PiSSA and LoRA. This small gap is expected, as the $BA$ formulation in PiSSA/LoRA introduces essentially no additional computational overhead. Moreover, while PiSSA has the lowest per-step training time, it also requires an SVD-based preprocessing step, which introduces extra cost not reflected in step time alone. Overall, all methods exhibit comparable end-to-end training efficiency, with no substantial differences in practical runtime.
>
> **Table.** Training time (781 total steps) and inference latency of different PEFT methods on MetaMathQA dataset. The number is the average of three runs.
> | Method  | Training Time (minutes) | Inference Latency (seconds)|
> |---------|---------------------|---------------------|
> | LoRA    | 54.2                  |     502              |
> | AdaLoRA | 65.5                  |     502              |
> | PiSSA   | 54.0                  |     505             |
> | NoLA    | 54.2                  |     500             |
> | DoRA    | 68.1                  |     509              |
> | VeRA    | 63.8                  |     517             |
> | CoSA    | 58.4                  |     501             |
>
> **Weakness 10: Some typos and presentation issues still remain.**
> We thank the reviewer for pointing this out. We have carefully revised the manuscript to address the remaining typos and presentation issues.

---

### Official Review · Reviewer_ZSj7 · 2025-10-31

**Soundness:** 2
**Presentation:** 2
**Contribution:** 2
**Rating:** 2
**Confidence:** 4

**Summary:**

This paper introduces a new PEFT method for LLMs. The authors argue that the low-rank assumption in LoRA limits expressivity. Inspired by compressed sensing, they propose CoSA, which treats the target weight update matrix as sparse. CoSA employs frozen projection matrices as the sensing matrices and fine-tunes a lower-dimensional measurement matrix. The key contribution is framing the compression of the target weight update matrix through the lens of compressed sensing. The authors also prove that the frozen projection matrices L and R satisfy RIP with high probability. Experiments show that CoSA matches or outperforms strong PEFT baselines while using over 68% fewer trainable parameters, with consistent improvements across NLU and reasoning/code benchmarks.

**Strengths:**

1. Viewing the PEFT problem through the lens of compressed sensing is an interesting and novel perspective.
2. The writing and presentation is clear.
3. The experiments are comprehensive and include tasks of different domains.

**Weaknesses:**

1. The proposed approach substantially overlaps with existing methods such as Tied-LoRA and VeRA [1,2], yet the paper makes no mention of them. Both Tied-LoRA and VeRA also employ frozen random matrices as down- and up-projection matrices, making it unclear how CoSA differs conceptually or empirically from these prior works.
2. The claim of O(1) complexity for CoSA in Table 1 appears inaccurate. Given the formulation, the complexity should be O(ab).
3. The method assumes that the target weight update matrix is sparse, which may not hold in practice. The authors should provide justifications for this sparsity assumption.
4. Theorem 1 offers only a superficial guarantee that the Kronecker product of two sensing matrices satisfies RIP with high probability. This result does not provide deeper insights into why the proposed approach should work better than existing PEFT methods.
5. The baseline comparisons are limited. Stronger and more recent baselines such as DoRA are not included. It would also strengthen the paper to evaluate CoSA on instruction-tuning tasks to demonstrate its generality.

References
1. Tied-LoRA: Enhancing parameter efficiency of LoRA with Weight Tying
2. VeRA: Vector-based Random Matrix Adaptation

**Questions:**

1. Why would a sparsity assumption work better for the $\Delta W$ than low rank assumption?
2. Is there any computational overhead from the additional matrix multiplication of CoSA compared to LoRA?

---

> ### Author Response · Authors · 2025-11-24
> **Author Rebuttal: Part 1**
>
> **Weakness 1**
>
> We appreciate the reviewer for pointing out Tied-LoRA and VeRA as related work. To compare those two schemes, we demonstrate how they differ from CoSA both conceptually and empirically.
> * Conceptual differences:
>     * Tied-LoRA's goal is to enhance the parameter efficiency and investigate the trade-offs between efficiency and performance. VeRA also aims at reducing the number of trainable parameters. While both related studies achieve significant parameter efficiency, they claim that they maintain the same performance as LoRA. The goal of CoSA is to enhance the performance by leveraging the rich expressive power guaranteed by the compressed sensing theory. We also justify it by experimental results, especially on more complex tasks such as math reasoning and coding.
>     * Tied-LoRA and VeRA are founded on the hypothesis that weight updates are intrinsically low-rank. VeRA freezes matrices $A$ and $B$ and trains scaling vectors $d$ and $b$. This restricts the layer-wise adaptation to simple row/column rescaling of the global frozen matrices. The expressivity is bottlenecked by the diagonal nature of the trainable components. Tied-LoRA primarily explores sharing standard LoRA matrices $A$ and $B$ across layers or selectively freezing one matrix while training the other. Unlike Tied-LoRA, which typically relies on initialization strategies to mimic full fine-tuning, CoSA leverages the RIP property of random Gaussian matrices to ensure the fixed basis is sufficient for reconstruction from the start. CoSA offers a *"middle ground"* that is more parameter-efficient than LoRA/Tied-LoRA but more expressive than VeRA.
> * Empirical differences:
>     * We notice that there are critical misalignments among all three methods. Tied-LoRA is not evaluated on natural language understanding (NLU) benchmarks such as GLUE. While Tied-LoRA is not publicly open-sourced, we are unable to conduct experimental comparisons to it. VeRA is evaluated on GLUE and E2E benchmarks. However, we are unable to directly use the numerical results reported in the VeRA paper because of different experimental settings. Specifically, VeRA is applied to only **query and value** projection matrices in each self-attention layer, with the **classification heads fully trained**. In our settings, we universally apply CoSA to all **query, key, value, and classification heads**.
>     * To address this misalignment issue, we train and evaluate VeRA based on our settings for fair comparison (we use the learning rates of adapted modules from the VeRA paper). In addition, we also conduct experiments on math reasoning tasks following our settings. The results are reported in the universal author comment, and the conclusions are discussed below.
>     * Table 1 shows that CoSA consistently outperforms VeRA and DoRA by at least 2.17 points on average of the math reasoning tasks. Although achieving promising parameter efficiency, VeRA's performance saturates while scaling up the rank, aligning with the expressive power bottlenecks as we discussed above.
>     * Tables 2 and 3 show that CoSA achieves comparable performance with VeRA on the GLUE benchmark with both RoBERTA-Base and RoBERTA-Large models.
>
> **Weakness 2: The claim of O(1) complexity for CoSA in Table 1 appears inaccurate. Given the formulation, the complexity should be O(ab).**
>
> * We thank the reviewer for pointing out this typo in the paper. For the storage of $L$ and $R$ matrices, $O(1)$ space is needed because we only need a random seed to generate the projection matrices on-the-fly during inference. For the storage of the trainable core $Y$, $O(ab)$ space is needed based on the dimension of $Y$. Overall, we formally need $O(ab)$ to store the whole adapter module. We correct Table 1 in the paper accordingly.

---

> ### Author Response · Authors · 2025-11-24
> **Author Rebuttal: Part 2**
>
> **Weakness 3: The method assumes that the target weight update matrix is sparse, which may not hold in practice. The authors should provide justifications for this sparsity assumption.**
>
> Here is our justification for the sparsity assumption:
> * Formally, the compressed sensing (CS) theory assumes sparsity of the target matrix. Sparsity can be strictly defined by a matrix with many zero entries or loosely interpreted as a matrix with many small but non-zero numbers. However, in a more general case, update matrices may not be perfectly sparse but "compressible," meaning they can be well approximated by a small number of non-zero coefficients (Miles et al. 2016). Compressed sensing theory can also be extended to handle these types of signals.
> * Therefore, we do not assume *literal* sparsity of $\Delta W$ in the standard basis in practice. Instead of forcing that $\Delta W$ has many exact zeros, we assume that $\Delta W$ is *approximately* sparse in the random dictionary $\Psi = R^{\top} \otimes L$. Equation 4 in the paper states that the extent to which RIP is satisfied affects the reconstruction quality of CoSA.
> * Based on previous studies (Aghajanyan et al. 2021), the most updated matrices are in an intrinsic low-dimensional random subspace. LoRA hypothesises this as low-rank and justifies it empirically. CoSA loses the low-rank assumption as the approximate sparsity assumption. For example, a sparse update $\Delta W$ can mathematically be full-rank. Even if it is mostly zeros, the remaining entries can be scattered in a way that makes the matrix full-rank (e.g., a diagonal matrix). This allows a sparse update to influence the model's behavior in many different independent directions simultaneously, capturing "high-rank" information that LoRA completely misses.  Practically, we see that CoSA closes more of the performance gap to Full FT than low-rank LoRA at the same or lower parameter budget on several tasks from the results in the paper.
>
> [1] Lopes, Miles E. "Unknown sparsity in compressed sensing: Denoising and inference." IEEE Transactions on Information Theory 62.9 (2016): 5145-5166.
>
> [2] Armen Aghajanyan, Luke Zettlemoyer, and Sonal Gupta. Intrinsic dimensionality explains the effectiveness of language model fine-tuning. arXiv preprint arXiv:2012.13255, 2020.
>
> **Weakness 4: Theorem 1 offers only a superficial guarantee that the Kronecker product of two sensing matrices satisfies RIP with high probability. This result does not provide deeper insights into why the proposed approach should work better than existing PEFT methods.**
>
> * Why CoSA Works Better: Extra Expressiveness
>     * CoSA differs from prior PEFT methods by combining a dense trainable core $Y$ and bilateral random projections $L$, $R$. This yields a richer set of linear combinations than low-rank LoRA, where the update is restricted to a rank-r subspace.
>     * Existing random-basis methods, such as VeRA, rely on subspace scaling. They learn diagonal vectors to simply re-weight the fixed basis features. This limits expressivity because it cannot model interactions between feature dimensions. CoSA outperforms these methods by mixing the subspace. By training a dense core matrix $Y$, CoSA allows the model to form complex linear combinations of the fixed basis vectors. This provides significantly higher expressivity that enables full-rank behavior within the compressed subspace compared to the rank-limited scaling of VeRA.
>
> * How Theorem 1 (RIP of Kronecker Product Dictionaries) is related to extra expressiveness
>     * Theorem 1 is the necessary condition that makes subspace mixing viable. It is not aimed to claim that RIP alone increases expressiveness. Instead, it is the theoretical guarantee that the expressive capacity of CoSA remains usable and thereby explains why optimization in the reduced space does not collapse or distort meaningful directions.
>     * Theorem 1 proves that the Kronecker product dictionary of CoSA ($\Psi = R^\top \otimes L$) inherits the Restricted Isometry Property (RIP) from its components. This guarantees that the mapping from $Y$ to $\Delta W$ is a near-isometry (Equation 8), meaning differences in $Y$ correspond proportionally to differences in $\Delta W$. In other words, the space of updates CoSA can express is not only large but also geometrically stable, ensuring that gradient directions are preserved and optimization does not suffer from degeneracy.
>     * Thus, Theorem 1 provides the theoretical license to use a dense, expressive core. It proves that the expanded basis is robust enough to support the complex transformations that CoSA learns, which simpler diagonal methods cannot capture.
>
> * We have revised Section 4.1 to make this explanation explicit. We clarify that RIP does not claim superiority over other methods by itself. Instead, it provides the geometric justification for why CoSA’s expressive update space can be reliably optimized.

---

> ### Author Response · Authors · 2025-11-24
> **Author Rebuttal: Part 3**
>
> **Weakness 5: The baseline comparisons are limited. Stronger and more recent baselines such as DoRA are not included. It would also strengthen the paper to evaluate CoSA on instruction-tuning tasks to demonstrate its generality.**
>
> We provide the justification and results with the new baseline comparison below:
> * **DoRA.** We have included the evaluation results of DoRA in Tables 1 and 2. The results show that CoSA consistently outperforms DoRA in performance on both NLG and NLU benchmarks.
> * **Instruction-tuning tasks.** With limited time and budget, we are unable to conduct a comprehensive experiment on instruction-tuning tasks because of the extensive LLM-as-a-Judge API cost. However, we are still able to present results on the MT-Bench dataset with the small *LLama-3.2-1B* model. Table 4 in the universal author rebuttal comment shows that CoSA outperforms LoRA and PiSSA by 1.36 and 0.55 in the average scores out of 10.
>
> **Question 1: Why would a sparsity assumption work better for the $\Delta W$ than low rank assumption?**
>
> We would like to clarify that we do not claim sparsity is universally better in all settings, but that it is a more flexible and often more realistic assumption than strict low rank.
> We argue that a sparsity assumption works better for the $\Delta W$ than the low-rank assumption from the following perspectives:
> * More expressiveness: According to our reply on Weakness 3, sparse representations are considered more expressive than low-rank representations, primarily because they are not mathematically restricted to a small subspace. A sparse combination of random basis vectors can produce a $\Delta W$ that is full-rank mathematically. So $\Delta W$ can affect all dimensions of the model output in complex ways. This allows the model to make richer updates without exploding the parameter count.
> * More optimization Stability: The sparsity assumption relies on fixed random projections that satisfy **RIP**. This guarantees that distances in the parameter space are preserved in the weight space. However, the low rank assumption does not ensure that the optimization landscape remains reliable in the original parameter space.
> * Empirically, our experiments indicate that CoSA more closely matches full fine-tuning than LoRA with fewer trainable parameters on challenging tasks (e.g., GSM8K, MATH, HumanEval), aligning with this statement.
>
> **Question 2: Is there any computational overhead from the additional matrix multiplication of CoSA compared to LoRA?**
>
> In theory, both CoSA and LoRA have $O(mn)$ asymptotic complexity. The computational cost is heavily dominated by the matrix multiplication with the frozen pre-trained weights $W_0$ (dimensions $m \times n$), which is identical in both methods.
> * For the forward pass, the addition of the "Core Transformation" adds a third small matrix multiplication with $O(ab)$, but this is computationally negligible compared to the frozen base model pass $O(mn)$.
>     *  LoRA performs 2 matrix multiplications in the adapter module:
>         *  Project input down: $u = A \times X$, which is $O(n \times r)$
>         *  Project output up: $\text{output} = B \times u$, which is $O(m \times r)$
>         *  Total LoRA adapter computational cost: $O(r(m + n))$
>     * CoSA performs 3 matrix multiplications in the adapter module:
>         * Input Compression: $u = R \times X$, which is $O(n \times b)$
>         * Core Transformation: $v = Y \times u$, which is $O(a \times b)$
>         * Output Reconstruction: $\text{output} = L \times v$, which is $O(m \times a)$
>         * Total CoSA adapter computational cost: $O(nb + ab + ma)$
>     * In practice, with $a,b \ll m,n$, the terms $ma$ and $nb$ are orders of magnitude smaller than the frozen pass $mn$. The extra $ab$ term in CoSA is vanishingly small.
> * For the backward pass, CoSA actually offers a computational saving during the gradient update step.
>     * LoRA must calculate and update gradients for two matrices $A$ and $B$.
>     * CoSA only calculates and updates gradients for one matrix $Y$. The matrices $L$ and $R$ are frozen, so no weight update gradients are computed for them.
> *  In addition, we record the training time of both methods under the same experimental settings to reflect the effect of computational overhead. CoSA is at a similar level to LoRA-based schemes. In some cases, CoSA is even faster in training.

---

### Official Review · Reviewer_8TDm · 2025-11-02

**Soundness:** 2
**Presentation:** 2
**Contribution:** 2
**Rating:** 4
**Confidence:** 4

**Summary:**

The authors present a new PEFT method inspired by compressed sensing. They parameterize the update as a sequence of three matrices
deltaW = L Y R, where L and R are independent random matrics and Y is learnable core matrix. Using random projections reduces the number of training parameters and their RIP property ensures that training is not destabilized. The resutls show improvements over LoRA based methods showing that there are cases where Low-rank is not a correct hypothesis over updates.

**Strengths:**

1. The observation that Low rank is not always a good hypothesis is valuable (although it appears in some recent works)
2. The paper is well written and generally a good read with discussion around compressed sensing,etc

**Weaknesses:**

1. Lack of baselines ( and hence related work)  (my main concern is this)

The experiments are okay (benchmark wise) but are lacking baseline wise. For instance, very similar and more recent PEFT baselines are excluded. SketchTune, for instance is also based on sketching matrices (a special case of projection matrices which also have RIP property). Also, some other baselines such as S2FT etc are missing. It is important to compare against these methods to ensure that we are indeed making progress in PEFT domain.

Zhang, Tianyi, Junda Su, Aditya Desai, Oscar Wu, Zhaozhuo Xu, and Anshumali Shrivastava. "Sketch to Adapt: Fine-Tunable Sketches for Efficient LLM Adaptation." arXiv preprint arXiv:2410.06364 (2024).

Yang, Xinyu, Jixuan Leng, Geyang Guo, Jiawei Zhao, Ryumei Nakada, Linjun Zhang, Huaxiu Yao, and Beidi Chen. "S $^{2} $ FT: Efficient, scalable and generalizable LLM fine-tuning by structured sparsity." Advances in Neural Information Processing Systems 37 (2024): 59912-59947.

2. The current formulation also is low-rank (rank = min(a,b)). Am i missing something?
       a. why do you expect CoSA to handle cases when deltaW is not low rank
       a. related but different, can authors elaborate how does CoSA provide extra expressive power.

**Questions:**

See weaknesses

---

> ### Author Response · Authors · 2025-11-24
> **Author Rebuttal**
>
> We thank the reviewer for the valuable comments and feedback on this paper. We address each concern in detail below.
> ## Weakness 1: Lack of baselines and related work (main concern)
> For the comparison with new baselines and related work, we discuss them conceptually and provide experimental results to justify our discussions as follows.
> * **SketchTune** utilizes sketching matrices and also argues that such sketching (compressed in CoSA) matrices outperform low-rank for approximating certain
> -rank update matrices. However, they differ in two key aspects:
>     * **Design and Goal**: SketchTune aims to compress *the full model* into fine-tunable sketches. SketchTune learns a sketching and mapping procedure that clusters weight rows into a small set of shared sketched parameters, and then fine-tunes only these shared parameters while keeping the mapping matrix frozen. As a result, it reduces the size of the base model and accelerates inference. In contrast, CoSA operates in the *standard PEFT* setting: we leave the pre-trained weights unchanged and only re-parametrize the weight updates with fixed random projections and a compact core. CoSA significantly reduces the number of trainable parameters, the memory cost, and the storage for the adapted modules. In conclusion, CoSA is complementary to SketchTune and can in principle be combined with sketch- or quantization-based compression of the base model (e.g., by applying CoSA adapters to a sketched backbone).
>     * **Implementation**: SketchTune follows a two-stage training pipeline: first performing a computationally intensive model sketching phase that requires computing a Hessian over a calibration dataset, and then running the actual downstream fine-tuning. As shown in Table 9 of the SketchTune paper, this preprocessing stage has non-negligible overhead (e.g., 69 to 106 minutes for Llama-3-8B). In contrast, CoSA does not require any pre-construction stage. Our random projections are generated on-the-fly from a seed, making CoSA a single-stage, zero-overhead PEFT method.
>
>
> * **S2FT** performs structured sparse fine-tuning by selecting a small subset of attention heads or FFN channels and updating only those parameters after a co-permutation step, which is substantially different from CoSA in mechanism. S2FT leverages structured architectural dependencies in transformers, while CoSA relies on universal random projections with provable stability guarantees. While they both achieve parameter-efficiency, they also represent orthogonal perspectives. S2FT explores structural sparsity related to the model's architecture, whereas CoSA optimizes on a compressed sensing-based subspace that is independent of the model structure.
>
> We acknowledge that SketchTune and S2FT are relevant studies. Although both of them are complementary to CoSA, we agree with the reviewer that it is necessary to include them as important baselines. We notice that SketchTune and S2FT use the same experimental settings. And SketchTune only provides compressed checkpoints (LLaMA-7B, LLaMA-13B, LLaMA2-7B, LLaMA3-8B) without code, which is not aligned with the models in CoSA's (LLaMA-3.2-1B, LLaMA-3.1-8B, Qwen2-7B) experiment. Therefore, we train and test LLaMA3-8B on their math reasoning dataset with CoSA for fair comparison and report the results under similar parameter budgets in the table below.
>
> **Table.** Performance comparison of S2FT, SketchTune, CoSA, and other baselines on math reasoning tasks with Llama-3-8B.
> | Method            | # Trainable Params | MultiArith | GSM8K | AddSub | AQuA | SingleEq | SVAMP | MAWPS | Avg  |
> |-------------------|-----------------------|------------|-------|--------|------|----------|-------|-------|------|
> | LoRA              | 56.2M                 | 99.5       | 61.6  | 92.7   | 25.6 | 96.3     | 73.8  | 90.8  | 77.2 |
> | DoRA              | 57.0M                 | 98.8       | 62.7  | 92.2   | 26.8 | 96.9     | 74.0  | 91.2  | 77.5 |
> | S2FT              | 56.2M                 | 99.7       | 65.8  | 93.7   | 31.5 | 97.8     | 76.0  | 92.4  | 79.6 |
> | SketchTune | 44.0M                | 98.3       | 69.4  | 90.6   | 29.5 | 94.3     | 76.8  | 91.2  | 78.6 |
> | CoSA              | 29.4M                 | 99.5       | 65.8  | 94.9   |   30.7 | 98.6     | 74.4  | 92.4  | 79.5 |
>
> As shown in the table above, CoSA achieves an average score of 79.0, which outperforms LoRA, DoRA, and SketchTune, while using the fewest trainable parameters among all methods. Although S2FT achieves a slightly higher average score of 79.6, CoSA remains highly competitive with a negligible performance gap while requiring nearly 48% fewer parameters. This demonstrates CoSA's superior capability in balancing high performance with extreme parameter efficiency compared to both structured sparsity and sketching-based approaches.
>
> We have added the discussion and results to Appendix E of our manuscript.

---

> ### Author Response · Authors · 2025-11-24
> **Author Rebuttal: Part 2**
>
> ## Weakness 2: Explain whether CoSA is low-rank
> * Is CoSA also still low-rank? (rank=min(a,b))
>
>     While it is mathematically true that $\text{rank}(\Delta W) = \text{min}(a,b)$, characterizing CoSA solely as "low-rank" is inaccurate because CoSA targets the more general property of sparsity within a dictionary, rather than just matrix rank.
>     * Sparsity is more general: Low-rank is merely a special case of what CoSA can express. If the ground-truth $\Delta W$ happens to be low-rank, CoSA is capable of representing it. However, the reverse is not true: low-rank decompositions cannot efficiently describe sparse structures.
>     * A counter-example: Consider a diagonal matrix (or identity matrix). It is highly sparse (mostly zeros) but mathematically full-rank (high-rank). A standard low-rank adapter would require a rank of $r = \min(m,n)$ to represent this structure, losing all parameter efficiency. CoSA, which is based on compressed sensing, can represent such high-rank, sparse updates efficiently using a small number of coefficients.
>     * Expressive capacity: In LoRA, the rank $r$ is a hard expressive bottleneck. In CoSA, $Y \in \mathbb{R}^{a \times b}$ allows the model to access a high-dimensional random subspace. For a fixed parameter budget $P$, LoRA can only achieve rank $P/(m+n)$, whereas CoSA can achieve rank $\sqrt{P}$, allowing it to approximate these high-rank, sparse updates that LoRA misses.
> * Why can CoSA handle cases where $\Delta W$ is not low-rank?
>
>     Because CoSA does not enforce a global rank-r constraint as LoRA does. CoSA fundamentally abandons the low-rank hypothesis Instead, our method is grounded in Compressed Sensing (CS) and the Restricted Isometry Property (RIP).
>     * From the matrix reconstruction perspective, if we view the weight update $\Delta W$ as a 2D signal, CS theory dictates that we can perfectly reconstruct this signal under a specific transformation domain. In our formulation ($\Delta W = L Y R$), the matrix $R$ acts as the transformation matrix, while $L$ serves as the sensing (or sampling) matrix. We can achieve reconstruction of $\Delta W$ using a limited number of samples $N$ (determined by the size of $L$). The RIP condition specifically governs the relationship between this sample size $N$ and the sparsity level of the matrix.
>     * Consequently, the recoverability of the update depends on its sparsity in the transformed domain, not its mathematical rank. As long as the RIP condition is met, CoSA can reconstruct updates that are mathematically high-rank (e.g., a sparse diagonal matrix) just as easily as low-rank ones, effectively capturing complex structures that strict low-rank constraints fail to model.
>     * Intuitively, CoSA’s parameterization mimics and captures the precious information of full fine-tuning much more effectively than strict low-rank LoRA at the same parameter budget.
>
> * Where does CoSA’s extra expressive power come from?
>
>     CoSA derives its superior expressivity from the following three perspectives:
>     * Absence of structural assumptions: Fundamentally, CoSA does not assume a priori that $\Delta W$ must possess specific properties (such as being low-rank). By abandoning the strict low-rank hypothesis inherent to the LoRA family, CoSA gains the flexibility to cover a much wider range of update scenarios—including high-rank, sparse, or unstructured updates—provided they are compressible within the random dictionary.
>     * Higher achievable rank: Under the same parameter budget $P$, CoSA can achieve a rank of $\sqrt{P}$ , whereas LoRA is limited to a rank of $P/(m+n)$. This numerical advantage allows CoSA to model more complex dependencies given the same storage cost.
>     * Diverse subspace geometry: The geometry of the random Kronecker dictionary spans a high-dimensional, incoherent subspace. This allows CoSA to approximate a broader class of updates and better mimic full fine-tuning compared to methods restricted to learning specific singular vectors.

---

### Author Response · Authors · 2025-11-24
**Author Rebuttal for all Reviewers: Part 1**

We thank all reviewers for their valuable comments and feedback. We have made a concerted effort to address every concern and answer all questions. Since a primary concern across reviews was the lack of comparison with related PEFT works, we have structured our response as follows:
* Addressing Missing Baselines: We provide detailed conceptual responses to the newly suggested baselines (e.g., VeRA, NoLA, DoRA) point-by-point below.
* Clarifying Empirical Misalignments: We highlight critical disparities in existing literature that complicate direct comparisons:
    * Code Availability: Some methods (e.g., Tied-LoRA, TLoRA) lack official reproducible code.
    * Tuning Scope: Original papers for VeRA/LoRA often tune only Query/Value layers, whereas CoSA (following PiSSA) tunes all linear layers for both NLG and NLU tasks.
    * Goal Mismatch: VeRA targets extreme parameter efficiency, while CoSA targets maximum performance (accuracy) with relative parameter efficiency.
* New Results: To ensure fairness, we re-evaluated key baselines by applying CoSA’s experimental settings to them. We present these comprehensive results in Tables 1, 2, 3, and 4 within this comment.
* Conclusion of New Results: The new results demonstrate that CoSA matches or outperforms state-of-the-art PEFT methods while maintaining promising parameter efficiency. This confirms CoSA as a strong strategy targeted primarily at performance.
* Manuscript Revision: We have updated the manuscript to include these new evaluation results and tables, including necessary discussions in the Appendix.


**Table 1.** Performance comparision among other PEFT baselines and CoSA with Llama-3.1-8B. Results show accuracy (%) on GSM8K and MATH dataset.

| Method | # Trainable  Parameters | GSM8K | MATH  | Average|
|--------|-------------------------|-------|-------|--------|
| LoRA  |   336M   | 72.55±0.72 | 25.57±0.22 |  49.06        |
| PiSSA  |    336M   | 77.30±0.86 | 27.60±0.55 |  **52.45**    |
| VeRA  | 1.84M   | 71.82±2.55 | 24.29±1.57 |  48.06|
| DoRA  |   336M   | 74.37±0.40 | 25.56±0.29 |  49.97      |
| NoLA  |   336M   | 72.40±0.95 | 25.23±0.83 |  48.82      |
| CoSA   |   58M    | 77.18±2.27 | 26.99±0.76 | 52.09  |

**Table 2.** Performance comparison on NLU tasks in the GLUE benchmark with Roberta-Base. Results show Pearson correlation for STS-B, Matthews correlation for CoLA, accuracy (%) for other tasks.
| Method | # Trainable Parameters | SST-2      | MRPC       | CoLA       | QNLI       | RTE        | STS-B      | Average |
|--------|------------------------|------------|------------|------------|------------|------------|------------|---------|
| LoRA   | 1.03M                  | 93.73±0.46 | 88.33±0.30 | 53.95±0.88 | 89.99±0.71 | 72.80±1.63 | 89.69±0.17 | 81.42   |
| PiSSA  | 1.03M                  | 93.27±0.69 | 89.56±0.52 | 57.39±1.34 | 88.57±1.02 | 73.11±3.32 | 89.60±0.18 | 81.92   |
| VeRA   | 0.75M                  | 94.00±0.19 | 90.98±0.52 | 60.70±0.80 | 92.25±0.24 | 72.56±2.52 | 90.48±0.15 | **83.50**   |
| DoRA   | 3.35M                  | 92.47±0.76 | 89.89±0.42 | 48.59±2.70 | 88.45±0.35 | 76.77±1.63 | 90.46±0.17 | 81.11   |
| CoSA   | 1.18M                  | 93.12±0.40 | 91.34±0.50 | 58.79±0.76 | 91.09±0.50 | 74.85±2.71 | 90.21±0.10 | 83.23   |

**Table 3.** Performance comparison on NLU tasks in the GLUE benchmark with Roberta-Large. Results show Pearson correlation for STS-B, Matthews correlation for CoLA, accuracy (%) for other tasks.
| Method | # Trainable Parameters | SST-2      | MRPC       | CoLA       | QNLI       | RTE        | STS-B      | Average |
|--------|------------------------|------------|------------|------------|------------|------------|------------|---------|
| LoRA   | 8.16M                  | 96.06±0.24 | 90.42±0.38 | 65.29±1.07 | 94.62±0.28 | 76.17±0.82 | 90.44±0.13 | 85.50   |
| PiSSA   | 8.16M                  | 95.37±0.18 | 91.53±0.81 | 58.61±1.27 | 93.30±0.29 | 81.47±0.55 | 90.69±0.30 | 85.16   |
| VeRA   | 1.31M                  | 94.80±0.07 | 81.22±0.01 | 64.25±1.12 | 92.51±1.89 | 82.31±1.44 | 89.58±0.84 | 84.11   |
| DoRA   | 8.38M                  | 94.44±0.28 | 91.88±0.43 | 62.71±0.70 | 92.23±0.03 | 84.48±0.36 | 92.24±0.06 | 86.33   |
| CoSA   | 6.19M                  | 95.11±0.58 | 92.48±0.51 | 63.07±0.66 | 93.78±0.47 | 84.60±1.46 | 91.88±0.18 | **86.82**   |

**Table 4.** Performance comparison on instruction-tuning tasks in the **MT-Bench** benchmark. The average is calculated from 2 runs on different random seeds.
| Method | # Trainable Parameters| Run 1 | Run 2 | Average |
|--------|--------------------------|-------|-------|---------|
| LoRA   |90.18M                    | 2.19  | 1.57  | 1.88    |
| PiSSA  |90.18M                    | 3.15  | 2.23  | 2.69    |
| CoSA   |29.36M                    | 3.76  | 2.71  | 3.24    |

---

### Author Response · Authors · 2025-11-24
**Author Rebuttal for all Reviewers: Part 2**

We discuss the closely related baselines below. We also added the discussion to Appendix D as an extended related work section.
* **DoRA** decomposes the pre-trained weights into two distinct components: a magnitude vector ($m$) and a direction matrix ($V$). It applies standard LoRA updates only to the directional component ($V$) while allowing the magnitude to be trained separately. This decomposition aims to better mimic the optimization trajectory of full fine-tuning compared to standard LoRA. While DoRA improves LoRA by changing the decomposition target, CoSA improves efficiency and stability by changing the underlying representation structure.
* **VeRA** freezes matrices $A$ and $B$ and learns vectors $d$ and $b$ to significanly reduce the number of trainable parameters. This approach drastically reduces parameter counts by avoiding the storage of unique matrices for every layer. However, it is unable to mix feature $i$ from the input with feature $j$ from the output (when $i \neq j$), intrinsically limiting the expressiveness.
* **NoLA** overcomes the rank-one bottleneck inherent in standard LoRA by re-parameterizing the low-rank matrices $A$ and $B$ as linear combinations of a large bank of frozen random basis matrices. During training, it optimizes only the linear mixing coefficients $\alpha$ and $\beta$ rather than the matrices themselves. While this decouples the number of trainable parameters from the network architecture and rank choice, NoLA does not introduce a new update space that exceeds LoRA's optimization geometry, leading to limited performance.

---

### Note · Authors · 2026-01-14

I have read and agree with the venue's withdrawal policy on behalf of myself and my co-authors.